# FaSTA*: Fast-Slow Toolpath Agent with Subroutine Mining for Efficient Multi-turn Image Editing

**Advait Gupta, Rishie Raj, Dang Nguyen**
University of Maryland, College Park
{advait25,rraj27,dangmn}@umd.edu

**Tianyi Zhou**
MBZUAI
tianyi.zhou@mbzuai.ac.ae

## Abstract

We develop a cost-efficient neurosymbolic agent to address challenging multi-turn image editing tasks such as "Detect the bench in the image while recoloring it to pink. Also, remove the cat for a clearer view and recolor the wall to yellow." It combines the fast, high-level subtask planning by large language models (LLMs) with the slow, accurate, tool-use, and local A* search per subtask to find a cost-efficient toolpath—a sequence of calls to AI tools. To save the cost of A* on similar subtasks, we perform inductive reasoning on previously successful toolpaths via LLMs to continuously extract/refine frequently used subroutines and reuse them as new tools for future tasks in an adaptive fast-slow planning, where the higher-level subroutines are explored first, and only when they fail, the low-level A* search is activated. The reusable symbolic subroutines considerably save exploration cost on the same types of subtasks applied to similar images, yielding a human-like fast-slow toolpath agent "FaSTA*": fast subtask planning followed by rule-based subroutine selection per subtask is attempted by LLMs at first, which is expected to cover most tasks, while slow A* search is only triggered for novel and challenging subtasks. By comparing with recent image editing approaches, we demonstrate FaSTA* is significantly more computationally efficient while remaining competitive with the state-of-the-art baseline in terms of success rate. Our code and data can be accessed here.

## 1 Introduction

Various practical applications require multi-turn image editing, which demands applying a sequence of heterogeneous operations—object detection, segmentation, inpainting, recoloring, and more—each ideally handled by a specialized AI tool. While it is challenging for existing text-to-image models (Rombach et al., 2022; Isola et al., 2018; Brooks et al., 2023; Zhang et al., 2024a) to tackle these long-horizon tasks directly, they can be broken down into a sequence of easier subtasks. Hence, a tool-using agent (Wang et al., 2024; Gao et al., 2024; Huang et al., 2023) has the potential to address each subtask by careful planning of a toolpath, i.e., a sequence of tool calls. Since AI tools' output quality and cost can vary drastically across different tasks and even samples, the planning often heavily depends on accurate estimation of the quality and cost of all applicable tools per step. This raises challenges to the exploration efficiency due to the great number of possible toolpaths and expensive computation of AI tools.

Large language model (LLM) agents usually excel at "fast" planning (Yao et al., 2023; Wei et al., 2023; Huang et al., 2022) that breaks down an image-editing task into a sequence of high-level subtasks without extensive exploration, thanks to their strong prior. However, they often misestimate AI tools' cost and quality due to a lack of up-to-date knowledge and the specific properties of each subtask. Moreover, they also suffer from hallucinations, e.g., choosing an expensive diffusion model instead of a simple filter when the latter suffices to complete the subtask. Compared to LLM agents, classical A* search requires "slow", expensive exploration on a dependency graph of tools, which in return can accurately produce an optimal, verifiable toolpath.

Can we combine the strengths of LLM agents in planning efficiency and A* search in editing accuracy to achieve an efficient, cost-sensitive solution for multi-turn image editing? A recent work,

Table 1: Top-5 frequently used subroutines extracted by inductive reasoning in FaSTA* from 100 CoSTA* toolpaths, ranked by their total selection frequency irrespective of the corresponding subtask.

| Subroutine | Subtask | Frequency |
|---|---|---|
| YOLO→ SAM→ SD Inpaint | Object Removal | 57/100 |
| Grounding DINO → SAM→ SD Inpaint | Object Recoloration | 49/100 |
| YOLO→ SAM→ SD Inpaint | Object Recoloration | 48/100 |
| YOLO → SAM→ SD Inpaint | Object Replacement | 25/100 |
| Grounding DINO→ SAM→ SD Erase | Object Removal | 22/100 |

CoSTA* (Gupta et al., 2025), proposes to leverage an LLM agent for a high-level subtask planning, producing a pruned subgraph of tools on which an A* search can be performed efficiently to find a cost-quality balanced toolpath. Despite its effectiveness in challenging image editing tasks and advantages over other baselines, the A* search remains a computational bottleneck.

Unlike humans who can learn reusable actions or create tools from past experiences and accumulate such knowledge over time, CoSTA* is a test-time only approach, so its exploration on previous tasks cannot be reused to improve or accelerate the planning for future tasks. In this paper, we study how to further reduce the cost of toolpath search by reusing the knowledge learned from explored tasks, a common feature of human learning. Inspired by photo editing applications that allow users to record their frequently used actions for future reuse, we propose to extract the repeatedly incurred subroutines of tool calls from the successful toolpaths of explored tasks. This is achieved automatically by performing inductive reasoning on LLMs given previous toolpaths. Each subroutine is represented by a symbolic rule under identifiable conditions, e.g.,

```
if object_area ≤ θ and mask_ratio > φ, then YOLO (Wang
et al., 2022)→ SAM (Kirillov et al., 2023)→ SD
Inpaint (Rombach et al., 2022) for Object Recoloration.
```

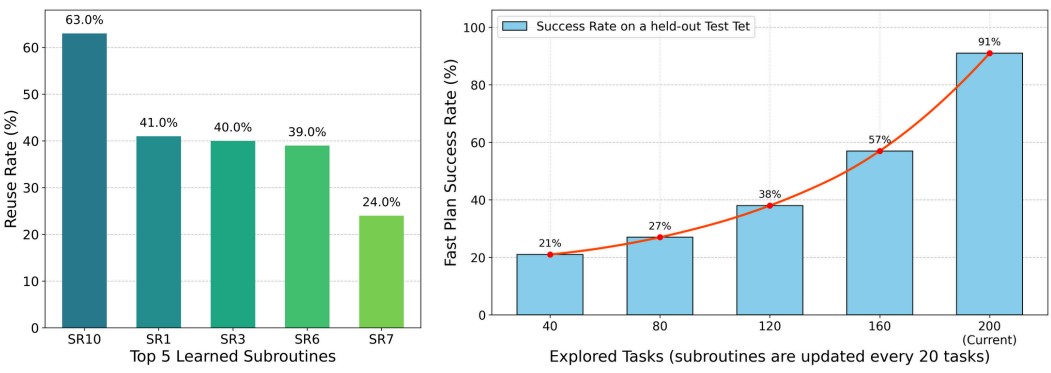

Figure 1: **Inductive Reasoning of Reusable Subroutines.** *Left*: Reuse rate (% of applicable subtasks where a subroutine was utilized) of the top-5 learned subroutines. *Right*: Success rate (%) of fast planning (subroutines only, without A* search) on subtasks for a held-out test set of tasks. It increases exponentially as more reusable subroutines are extracted from an increasing number of explored tasks.

Table 1 summarizes the top five subroutines extracted by LLMs from CoSTA* toolpaths on 100 tasks, while Figure 1 illustrates their reuse rates. They show an unexplored reusability of subroutines for image editing tasks. Motivated by this observation, we propose *a neurosymbolic LLM agent with a learnable memory*, "Fast-Slow Toolpath Agent (FaSTA*)", that keeps mining symbolic subroutines from previous experiences and reuses them in the exploration and planning for future tasks. Its exploration stage can be explained as a novel and critical improvement to existing in-context reinforcement learning (ICRL): instead of recording all previous tools (Wang et al., 2022; Kirillov et al., 2023; Rombach et al., 2022; Liu et al., 2024) and paths and drawing contexts from them

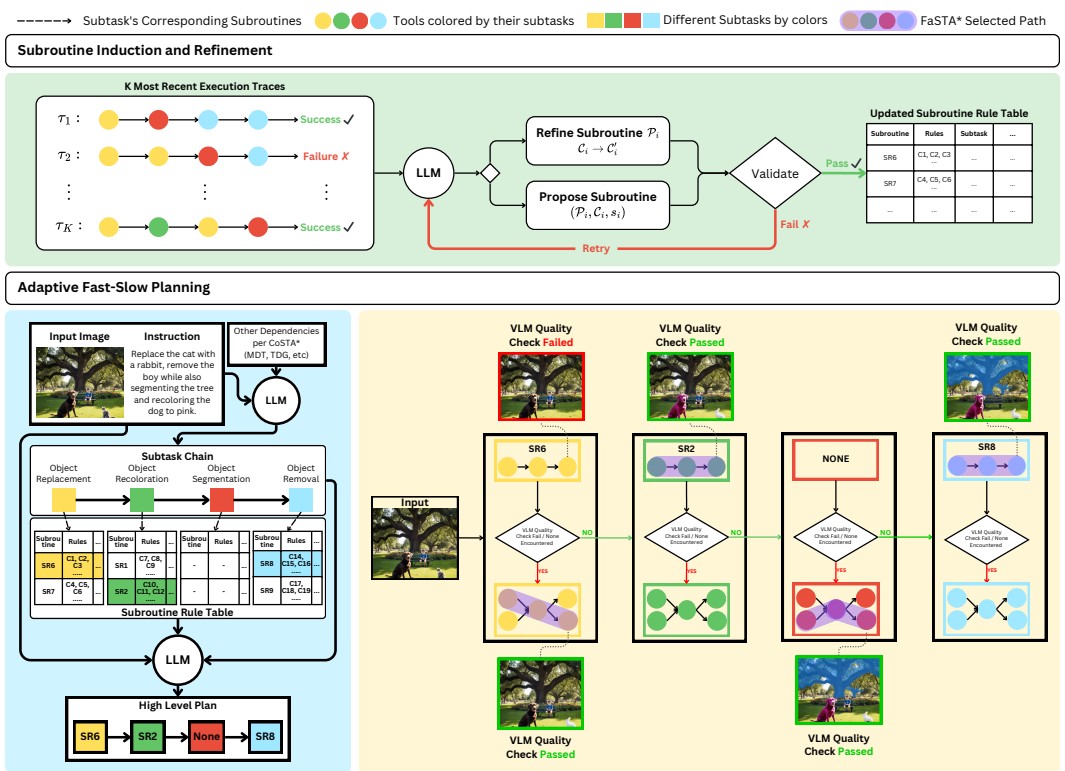

Figure 2: *Top:* **Online learning (induction) and refinement of reusable subroutines** from explored toolpaths for previous tasks. *Bottom:* **Adaptive fast-slow planning framework in FaSTA\*.** Given a new task, FaSTA\* first uses an LLM to generate a high-level plan of subtasks and then select a subroutine per subtask, yielding a fast plan. Only when the subroutine's output does not pass the quality check by VLMs, a slow planning by A\* search on the subtask's tool subgraph will produce a toolpath for the subtask.

for future exploration, FaSTA\* condenses these paths to a few reusable subroutines by inductive reasoning and uses these principles to guide future tasks more effectively.

As an imitation of the fast-slow planning in human cognition, the reusable subroutines enable "fast planning" of FaSTA\* in the planning stage. In addition to the subtask-level planning in CoSTA\*, the LLM agent in FaSTA\* further chooses a subroutine for each subtask. Only when there does not exist any subroutine for the subtask, or when the subroutine's output fails to pass the quality check by VLMs, FaSTA\* will resort to the "slow planning", i.e., A\* search on a low-level subgraph of tools for the subtask. Hence, FaSTA\* can entirely avoid the expensive low-level A\* search if the fast plan succeeds. Moreover, as more tasks have been explored and more subroutines collected, most incoming tasks can be handled by fast planning, and only very unique or rare tasks require slow planning. Our main contributions can be summarized as:

1. **A memory of Symbolic Subroutine learned by LLMs:** We extract reusable subroutines as symbolic rules from explored tasks' toolpaths by inductive reasoning on LLMs. They serve as high-level actions and significantly reduce the exploration cost for future tasks.

2. **Fast-Slow Planning of Toolpaths:** We develop a neurosymbolic toolpath agent FaSTA\* that benefits from fast planning (LLMs' subtask planning and selection of symbolic subroutines) to produce a toolpath efficiently, and only invokes slow A\* search when the fast planning fails.

3. FaSTA\* achieves better cost–quality trade-offs across diverse tasks than most baselines. Compared to CoSTA\*, it can save the cost by **49.3%** with the price of merely **3.2%** quality degradation.

## 2    RELATED WORK

**Multi-turn Image Editing.**    Generative models excel at image synthesis and single-edits (Sordo et al., 2025; Saharia et al., 2022; Nichol et al., 2022; Hertz et al., 2022), (e.g., (Rombach et al., 2022); (Ramesh et al., 2021); (Isola et al., 2018); (Chen et al., 2023a)), but multi-turn editing from composite instructions presents unique challenges. Controllability techniques (e.g., ControlNet (Zhang et al., 2023), sketch-guidance (Voynov et al., 2022), layout-synthesis (Chen et al., 2023b; Li et al., 2023)) improve short-sequence precision. Recent unified multimodal models, such as BLIP-3o (Chen et al., 2025a) and BLIP3o-NEXT (Chen et al., 2025b), have shown promise by tackling generation and editing natively within a single autoregressive and diffusion architecture. However, iteratively applying standard models like InstructPix2Pix (Brooks et al., 2023) or those from MagicBrush (Zhang et al., 2024a) often lacks long-sequence coherence without sophisticated planning. Agentic systems (e.g., GenArtist (Wang et al., 2024), CLOVA (Gao et al., 2024)) decompose tasks for tool use. Nevertheless, efficient execution, especially for recurrent operation patterns, is a key challenge, with a lack of learning from past patterns causing redundant computational effort.

**Tool-use Agent and Planning.**    LLMs as reasoning agents coordinating tools show broad impact (Zhang et al., 2024b; Gou et al., 2024; Qu et al., 2025; Li, 2024), especially in image editing for orchestrating specialized tools (Gupta et al., 2025; Gao et al., 2024; Wang et al., 2024; Shen et al., 2024). Closely related to our approach, MuLan (Li et al., 2024) uses LLM decomposition and iterative VLM feedback to progressively generate complex multi-object images. MLLM agents (e.g., CoSTA* (Gupta et al., 2025), GenArtist (Wang et al., 2024), SmartEdit (Huang et al., 2023)) plan by decomposing goals; yet, others like *Visual ChatGPT* (Wu et al., 2023) and MM-REACT (Yang et al., 2023) often explore tool calls greedily, ignoring budgets, while dialog systems enable iterative refinement (Huang et al., 2024a). Agent planning often leverages LLM emergent reasoning (Yao et al., 2023; Huang et al., 2022; 2024b; Gupta & Kembhavi, 2023), sometimes augmented by explicit reasoning steps (Yao et al., 2023; Wei et al., 2023). However, generating efficient, optimal low-level tool sequences for complex image editing is challenging, as LLMs may falter without further guidance or search (Huang et al., 2024b; Fu et al., 2024). CoSTA* (Gupta et al., 2025) improves this with LLM-planned A* search, but intensive search costs persist without experience reuse. Robust learning and symbolic reuse of common, successful tool sequences are largely missing, hindering efforts to reduce planning costs and build scalable, adaptive image editing agents.

## 3    PRELIMINARIES: CoSTA* AND EFFICIENT TOOLPATH SEARCH

We build FaSTA* based upon a recent framework, CoSTA* (Gupta et al., 2025), to find cost-efficient toolpaths for multi-turn image editing. CoSTA* uses an LLM agent to prune a high-level subtask graph on which a low-level A* search is performed to determine the final toolpath. We provide a brief overview of CoSTA* below. More comprehensive details are available in Appendix E.

**Foundational Components of CoSTA*.**    CoSTA* utilizes three pre-defined knowledge structures:
- A **Tool Dependency Graph (TDG)** to map prerequisite input/output dependency relationships between AI tools. It can be automatically generated or human-crafted.
- A **Model Description Table (MDT)** to catalog AI tools, their supported subtasks (e.g., object detection, recoloration), and input/output.
- A **Benchmark Table (BT)** with cost/quality data for (subtask, tool) pairs collected from published works. It is used to initialize the heuristics in A* search.

These elements collectively enable mapping from high-level task requirements to specific tool sequences and their predicted performance.

**CoSTA* Planning Stages.**    The planning in CoSTA* unfolds in three main stages:

1. **Subtask-Tree Generation:** An LLM interprets the user's natural language instruction and input image to produce a *subtask tree*, $G_{ss}$. This tree decomposes the overall task into smaller subtasks. CoSTA* allowed this tree to have parallel branches representing alternative subtask plans.[1]

---

[1] We found that a single branch is sufficient in practice and reduces CoSTA*'s complexity (see Appendix F).

2. **Tool Subgraph Construction:** The subtask tree is then used to identify relevant tools from the MDT and their dependencies from the TDG. This forms a final *Tool Subgraph* for each task, $G_{ts}$, which contains all tools for all subtasks required to accomplish the final editing task.

3. **Cost-Sensitive A$^*$ Search:** CoSTA$^*$ performs an A$^*$ search on $G_{ts}$ to find an optimal toolpath. The search is guided by a cost function $f(x) = g(x) + h(x)$, where $g(x)$ is the accumulated actual cost (considering time and VLM-validated quality) and $h(x)$ is the estimated heuristic value (derived from the BT), estimating the cost-to-go till leaf node. VLM checks after tool executions allow for dynamic updates and path corrections upon failure.

## 4 FAST-SLOW TOOLPATH AGENT (FASTA$^*$)

### 4.1 FROM COSTA$^*$ TO FASTA$^*$

While CoSTA$^*$ (Gupta et al., 2025) offers a solid foundation for tackling multi-turn image editing, its reliance on A$^*$ search, especially for complex tasks with large Tool Subgraphs, can be computationally intensive. Moreover, CoSTA$^*$ is a test-time planning method that cannot learn from existing experiences to accelerate future tasks' planning. This may lead to inefficient, repeated A$^*$ search of the same subroutines, given our observation of many recurring ones across tasks.

To further reduce the cost of CoSTA$^*$ on A$^*$ search and avoid exploring the same subroutines repeatedly, we equip CoSTA$^*$'s hierarchical planning with online learning of symbolic subroutines frequently used in explored tasks' toolpaths, and choose from them to address similar subtasks in later tasks, resulting in a novel, efficient In-Context Reinforcement Learning (ICRL) (Monea et al., 2025) framework. FaSTA$^*$ still follow CoSTA$^*$'s initial step of decomposing each task into a chain of subtasks, but saves considerable computation by a novel fast-slow planning that lazily triggers A$^*$ search for each subtask only when the selected subroutine fails.

First, we introduce a system for **online inductive reasoning of subroutines** (Section 4.2), where FaSTA$^*$ learns from past successful (and unsuccessful) editing experiences. It identifies frequently used, effective subsequences of tool calls for subtasks (subroutines) and the general conditions under which they perform well. Second, these learned subroutines form the backbone of an **adaptive fast-slow execution strategy** (Section 4.3). Instead of immediately resorting to a detailed A$^*$ search, FaSTA$^*$ first attempts to apply a "Fast Plan" composed of these proven subroutines. If this fast plan is unsuitable for a subtask or fails a quality check, FaSTA$^*$ then dynamically engages a more meticulous "slow planning" of tool calls for each subtask by A$^*$ search.

This fast-slow planning allows FaSTA$^*$ to handle many tasks rapidly by selecting from learned subroutines. However, it retains the ability to perform deeper, more complex searches when faced with novel or challenging scenarios. The goal is to achieve a better balance of computational cost and high-quality, making complex image editing more practical and efficient.

### 4.2 ONLINE LEARNING AND REFINEMENT OF REUSABLE SUBROUTINES

Our first core contribution is a neurosymbolic method for automatically discovering and refining reusable subroutines from execution data (i.e., traces – representing the detailed execution data logged for each subtask) during online operation. A subroutine $\mathcal{P}_s = (t_1, t_2, \ldots, t_k)$ represents a frequently observed, ordered sequence of tool calls $t_i \in V_{td}$ that effectively accomplish a specific subtask $s$ under certain conditions $\mathcal{C}_s$. The goal is to learn and maintain a dynamic library of rules $\mathcal{R} = \{(\mathcal{P}_j, \mathcal{C}_j, s_j)\}_{j=1}^M$ mapping subtasks and context features to cost-effective subroutines, stored in a Subroutine Rule Table (see Appendix I for the structure).

Our approach diverges from standard ICRL, which often grapples with cumbersome raw experience logs and inefficient generalization. FaSTA$^*$ employs an LLM for an **explicit inductive reasoning step**, analyzing execution traces not just to sample past experiences, but to synthesize compact, symbolic (subroutine, activation rule, subtask) knowledge leading to more interpretable and generalizable rules. *Crucially, to ensure the generalizability of subroutines and a fair evaluation, the inductive reasoning of subroutines is performed on a held-out set of new diverse tasks (e.g., random internet images with new complex prompts) excluded from the benchmark.* This online learning of symbolic rules and subroutines aims to create an interpretable and off-the-shelf library of action rules $\mathcal{R}$ that can

be periodically augmented and refined based on new experiences (Algorithm 1, Appendix G). The online learning and adaptation cycle in FaSTA* involves the following key stages:

1. **Data Logging:** FaSTA* continuously records detailed execution data $\tau$ (or traces) from each subtask. The motivation is to capture rich, contextualized data about what paths were taken, under what conditions (e.g., object size from YOLO, mask details from SAM, LLM-inferred context like background complexity), and with what outcomes (cost, quality, failures). This logged data serves as the raw experiences for subroutine learning. Full details are provided in Appendix M.

2. **Periodic Refinement:** To balance continuous learning with operational stability, the refinement process is triggered periodically (every $K = 20$ tasks), using the most recent batch of accumulated traces ($\mathcal{T}_{recent}$). This ensures the system adapts to evolving patterns without excessive computational overhead.

3. **Inductive Reasoning by LLM:** This is the core knowledge synthesis step. The LLM is prompted with $\mathcal{T}_{recent}$ and the current rule set $\mathcal{R}$ to perform inductive reasoning, which analyzes these experiences, identifies recurring successful subroutines, and infers the contextual conditions (activation rules $\mathcal{C}_j$) determined via robust semantic bucketing instead of numerical values under which they are effective, or proposes modifications to existing rules. This allows FaSTA* to generate new hypotheses about efficient strategies. Appendix S provided the detailed prompts.

4. **Verification and Selection of Subroutines:** Recognizing LLM-proposed rules as hypotheses, this stage rigorously validates each change $\Delta$ before integrate it into $\mathcal{R}$ to ensure beneficial, robust subroutines and prevent degradation from flawed rules. Validation uses specialized test datasets against a baseline (CoSTA* or current FaSTA*). A "Net Benefit" score (balancing cost/quality) determines acceptance, with an LLM-based retry mechanism for refinement. Further details and evaluation protocols are provided in Appendix G.4 and Appendix L, respectively.

More details of inductive reasoning can be found in Appendix G. This online adaptation loop allows FaSTA* to continuously learn from its operational data, building and refining a library of cost-effective, validated subroutines that enhance its "fast planning" capabilities. Figure 1 shows the reuse rate of learned subroutines and how they improve the success rate of fast planning with increasing experiences. More details are provided in Appendix H.

## 4.3 Adaptive Fast-Slow Planning

Following the generation of the subtask chain (Section 3) and the online refinement of the subroutine rule library $\mathcal{R}$ (Section 4.2), FaSTA* employs its second main contribution: an **adaptive fast-slow planning and execution strategy**. This approach aims to drastically reduce execution cost by prioritizing an efficient "fast plan" of learned subroutines, while retaining the robustness of A* search for novel or failed subtasks via a localized "slow plan" fallback. Figure 2 illustrates this process, and Algorithm P.2 details the execution flow. The process involves two key phases: fast plan generation and the adaptive fast-slow execution.

Table 2: Adaptive Fast-Slow Planning Fallback Statistics. It shows the percentage of subtasks that can be addressed by subroutines vs. those requiring fallback to low-level A* search.

| Execution Path | Percentage |
|---|---|
| High-Level (Subroutine Success) | 91% |
| Low-Level Fallback (No Subroutine or Check Fails) | 9% |

**1. Fast Planning.** Initially, FaSTA* generates a high-level "Fast Plan" $\mathcal{M}_{subseq}$ without search. An LLM (GPT-4o, with prompt in Appendix R) takes an input image, user prompt, subtask plan $s_{1:N}$, and the subroutine rule set $\mathcal{R}$ to select an optimal subroutine $\mathcal{P}_{s_i} \in \mathcal{R}$ or "None" for each subtask $s_i$. The selection considers the activation rules $\mathcal{C}_j$ associated with each potential subroutine $\mathcal{P}_j$ and the current image context. If the context satisfies Bi-Directional Block Self-Attention for Fast and Memory-Efficient Sequence Modeling activation rules for multiple subroutines $\mathcal{P}_j$ applicable to $s_i$, the one minimizing the cost-quality tradeoff score $C_{\text{avg}}(\mathcal{P}_j)^\alpha \times (2 - Q_{\text{avg}}(\mathcal{P}_j))^{2-\alpha}$ is chosen, where $C_{\text{avg}}(\mathcal{P}_j)$ and $Q_{\text{avg}}(\mathcal{P}_j)$ are the averaged cost and quality of subroutine $\mathcal{P}_j$ in different existing

toolpaths, and $\alpha$ is a used-defined trade-off coefficient as in CoSTA$^*$ (Gupta et al., 2025). This process yields the fast plan $\mathcal{M}_{subseq} = (\mathcal{P}_{s_1}, \ldots, \mathcal{P}_{s_N})$.

**2. Adaptive Fast-Slow Execution.** The fast plan $\mathcal{M}_{subseq}$ is then executed sequentially.

- **Fast Plan Attempt:** For each planned subtask $s_i$, FaSTA$^*$ calls the tools within $\mathcal{P}_{s_i}$ sequentially and check the quality of each tool execution using a VLM as in CoSTA$^*$ ((Gupta et al., 2025, Appendix I, Fig. 12) lists details regarding the VLM check criteria).
- **Slow Planning Trigger:** If the fast plan cannot find any subroutine for $s_i$ (i.e., $s_i$ =None), or if any tool execution in $\mathcal{P}_{s_i}$ fails to pass the quality check, FaSTA$^*$ switches to the "slow planning" for the current subtask $s_i$. Hereby, the detailed low-level subgraph $G_{low}(s_i)$ for $s_i$ is first constructed as per CoSTA$^*$ (Gupta et al., 2025, Sec. 4.2). Subsequently, an A$^*$ search is performed on this $G_{low}(s_i)$ to find an optimal path to complete $s_i$, employing the same cost function and search algorithm as defined in CoSTA$^*$ (Gupta et al., 2025, Sec. 4.3–4.5).

This adaptive fast-slow planning ensures that the more efficient fast plan is executed by default. In contrast, the robust but more expensive slow planning by A$^*$ is invoked lazily only for those subtasks where the fast plan fails. The fallback statistics of fast-slow planning is reported in Table 2.

## 5 EXPERIMENTS

We conduct extensive experiments to evaluate the effectiveness of FaSTA$^*$. We aim to answer: (1) How does our method compare to the state-of-the-art CoSTA$^*$ in terms of execution cost and output quality? (2) How robust is the fast-slow planning approach, and what is the impact of its core components? All experiments have been conducted on a single NVIDIA A100 GPU.

### 5.1 EXPERIMENTAL SETUP

**Dataset.** We evaluate our method on the benchmark dataset curated and released alongside CoSTA$^*$ ( (Gupta et al., 2025)). This dataset comprises 121 image-prompt pairs, featuring complex multi-turn editing instructions involving 1-8 subtasks per image (amounting to 550 total image manipulations or turns across the dataset), covering both image-only and mixed image-and-text manipulations (Gupta et al., 2025, Appendix D). Its diversity and complexity make it suitable for assessing the cost-saving potential and robustness of our adaptive planner. To further ensure generalizability, we also evaluate FaSTA$^*$ on the Complex-Edit benchmark, with results in Table 4

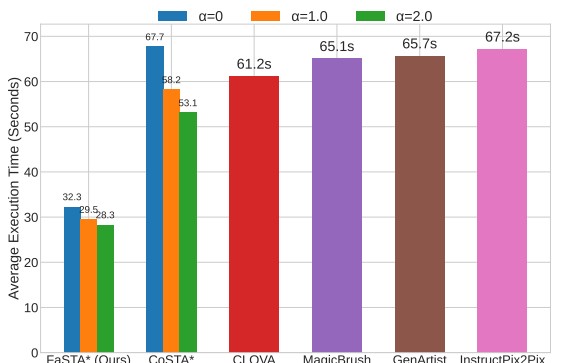

Figure 3: Execution time (seconds) per image. FaSTA$^*$ and CoSTA$^*$ costs vary with tradeoff coefficient $\alpha$. Baseline costs are from CoSTA$^*$ (Gupta et al., 2025).

**Baselines.** Our primary baseline is the original **CoSTA$^*$** algorithm (Gupta et al., 2025), which represents the state-of-the-art cost-sensitive planner using A$^*$ search on a pruned tool subgraph. In our ablation studies and discussions, we refer to this as the "Low-Level Only" approach. We also acknowledge other agentic and non-agentic baselines evaluated in the CoSTA$^*$ paper (e.g., GenArtist (Wang et al., 2024), CLOVA (Gao et al., 2024), InstructPix2Pix (Brooks et al., 2023), MagicBrush (Zhang et al., 2024a)), primarily to contextualize the Pareto optimality results, while noting their limitations in handling the full range of tasks or performing cost-sensitive optimization.

**Evaluation Metrics.** We adopt the evaluation scheme from CoSTA$^*$ (Gupta et al., 2025):

- **Quality (Human Evaluation):** Task success is measured by human evaluation. Following (Gupta et al., 2025), each subtask $s_i$ within a task $T$ is assigned a score $A(s_i) \in \{0, x, 1\}$, where $x \in (0, 1)$ represents partial correctness based on predefined rules. The task accuracy $A(T)$ is the average of its subtask scores, and the overall accuracy is the average $A(T)$ across the dataset (Gupta et al.,

Table 3: Detailed Quality Comparison (Average Human Evaluation Score) across Task Types and Complexities. Baseline results are from CoSTA* (Gupta et al., 2025). Both FaSTA* and CoSTA* adopt a balanced cost-quality trade-off by setting $\alpha = 1$.

| Task Type | Task Category | FaSTA* (Ours) | CoSTA* | VisProg | CLOVA | GenArtist | Instruct Pix2Pix | MagicBrush |
|---|---|---|---|---|---|---|---|---|
| Image-Only Tasks | 1-2 subtasks | 0.92 | **0.94** | 0.88 | 0.91 | 0.93 | 0.87 | 0.92 |
| | 3-4 subtasks | 0.91 | **0.93** | 0.76 | 0.77 | 0.85 | 0.74 | 0.78 |
| | 5-6 subtasks | 0.92 | **0.93** | 0.62 | 0.63 | 0.71 | 0.55 | 0.51 |
| | 7-8 subtasks | 0.91 | **0.95** | 0.46 | 0.45 | 0.61 | 0.38 | 0.46 |
| Text+Image Tasks | 2-3 subtasks | 0.91 | **0.93** | 0.61 | 0.63 | 0.67 | 0.48 | 0.62 |
| | 4-5 subtasks | 0.92 | **0.94** | 0.50 | 0.51 | 0.61 | 0.42 | 0.40 |
| | 6-8 subtasks | 0.91 | **0.94** | 0.38 | 0.36 | 0.56 | 0.31 | 0.26 |
| Overall Accuracy | Image Tasks | 0.91 | **0.94** | 0.69 | 0.70 | 0.78 | 0.64 | 0.67 |
| | Text+Image Tasks | 0.91 | **0.93** | 0.49 | 0.50 | 0.61 | 0.40 | 0.43 |
| | All Tasks | 0.91 | **0.94** | 0.62 | 0.63 | 0.73 | 0.56 | 0.59 |

2025). We rely on human evaluation due to the limitations of automated metrics like CLIP (Radford et al., 2021) in capturing nuanced errors in complex, multi-step, multimodal editing tasks. More details about the reasons for resorting to human evaluation can be found in Sec. 5.2 of (Gupta et al., 2025). More details about the evaluation process and rules for assigning partial scores are mentioned in Appendix K.

- **Cost (Execution Time):** Efficiency is measured by the total execution time in seconds for each inference, including any necessary retries.

## 5.2 MAIN RESULTS

**Performance Analysis.** As shown in Table 3 and Figure 3, FaSTA* achieves average quality remarkably close to the original CoSTA* method, with only a minimal drop of approximately 3.2% in accuracy. However, the benefits in efficiency are substantial. Our approach reduces the average execution cost by over 49.3%, achieving costs nearly half of CoSTA*. This demonstrates the effectiveness of the adaptive slow-fast strategy: by leveraging learned subroutines for common cases ("fast planning"), we drastically cut down on expensive A* exploration within low-level tool subgraphs, while the

Table 4: Performance comparison on subset of Complex-Edit benchmark. FaSTA* achieves consistent cost reductions while maintaining quality across varying task complexities.

| Task Complexity | Metric | CoSTA* | FaSTA* (Ours) |
|---|---|---|---|
| 1-3 Subtasks | Cost (s) | 46.75 | **35.87** |
| | Accuracy | 0.88 | 0.86 |
| 4-5 Subtasks | Cost (s) | 77.25 | **54.17** |
| | Accuracy | 0.90 | 0.89 |
| 6-8 Subtasks | Cost (s) | 105.60 | **73.20** |
| | Accuracy | 0.90 | 0.88 |
| **Overall** | **Avg. Cost (s)** | 78.27 | **55.12** |
| | **Avg. Accuracy** | **0.89** | **0.87** |

fallback mechanism ("slow planning") ensures that quality is not significantly compromised when subroutines are unsuitable or fail. Confirming these gains generalize, evaluations on the independent Complex-Edit benchmark (Table 4) demonstrate a ∼30% cost reduction with comparable quality. More details on quantitative results on comparisons with other methodologies and models are available in Appendix J.

**Pareto Optimality Analysis.** Figure 5 illustrates the Pareto frontiers achieved by varying the cost-quality tradeoff coefficient $\alpha$. FaSTA* consistently achieves a superior frontier compared to CoSTA* (Low-Level Only) and dominates the other baselines evaluated in (Gupta et al., 2025). For any given quality level achieved by CoSTA*, our method offers a significantly lower cost, and for any given cost budget, our method yields comparable or slightly lower quality. This highlights the adaptability of our approach in catering to different user preferences regarding the balance between execution speed and output fidelity, while consistently operating at a more efficient frontier than searching the low-level graph alone. Like we had already

Figure 5: **Cost-Quality Pareto Frontier.** FaSTA* with various $\alpha$ values against CoSTA* and other baselines. FaSTA* achieves a superior frontier, offering better cost-quality trade-offs.

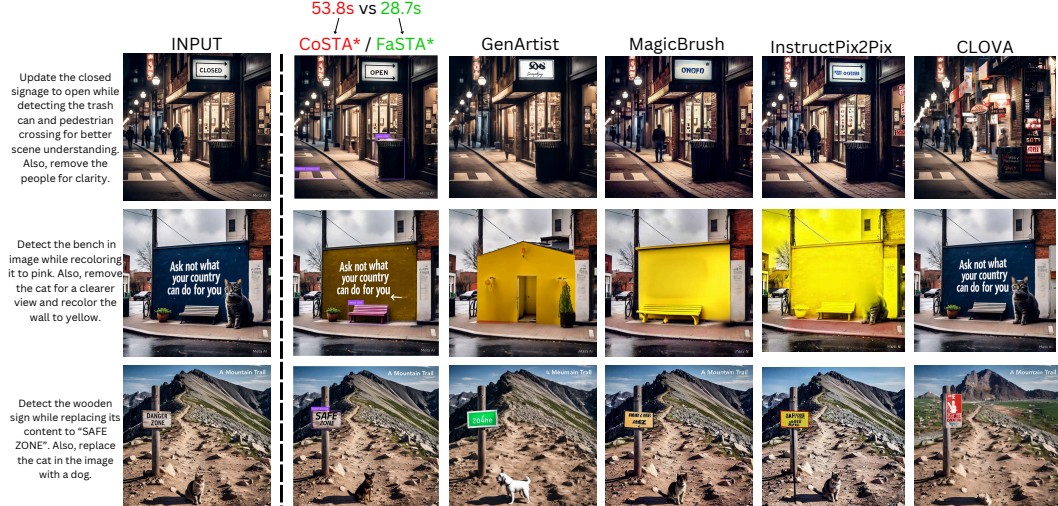

Figure 4: **Comparison of FaSTA\*** with CoSTA\* (Gupta et al., 2025) and other leading image editing agents for complex multi-turn tasks. FaSTA\* achieves visual results identical to CoSTA\* and significantly surpasses other baselines in accuracy and coherence. Notably, FaSTA\* delivers this high quality at roughly half the execution cost of CoSTA\*, highlighting its superior efficiency.

mentioned earlier, FaSTA\* becomes increasingly optimal with more subroutine experience. This cost-quality tradeoff compared to CoSTA\* w.r.t. number of subroutines mined is explained in Appendix C.1 in greater detail.

**Qualitative Analysis** Figure 4 provides qualitative comparisons of FaSTA\* against baselines, illustrating its ability to handle complex multi-turn editing tasks effectively. These examples showcase the practical benefits of our adaptive fast-slow approach in achieving desired edits. For a more detailed qualitative analysis, including specific case studies demonstrating subroutine effectiveness, and comparisons of different tool paths under specific conditions, please refer to Appendix C. We also do a similar qualitative comparison of FaSTA\* with the very recent **Gemini 2.0 Flash Preview Image Generation** on some tasks from the CoSTA\* benchmark dataset. These results can be seen in Figure 14.

## 5.3 Ablation Studies

**Impact of Subroutine Verification.** To evaluate the significance of our subroutine verification step (Section 4.2, Step 4), we compared FaSTA\*'s performance with and without this validation. We first used an LLM to propose subroutine changes based on initial task traces. These proposals were then either inducted directly or only after passing our full verification process. The outcomes are in Table 5. The verification step proved crucial: FaSTA\* with verified subroutines yielded much fewer fallbacks to low-level A\* search. This improvement stems from ensuring only reliable subroutines are adopted, which prevents wasted execution on flawed proposals and avoids misdirecting the agent when low-level search is genuinely needed.

Table 5: (Left) Impact of Subroutine Verification; (Right) Fast/Slow planning only vs. FaSTA\*

| Method | Low-Level Fallback (%) |
| --- | --- |
| FaSTA\* (w/o Verification) | 28% |
| FaSTA\* (w/ Verification) | 9% |

| Method | Avg. Quality | Avg. Cost (s) |
| --- | --- | --- |
| Fast plan Only | 0.84 | 27.5 |
| Slow plan Only | 0.93 | 46.8 |
| FaSTA\* (Ours) | 0.91 | 29.5 |

**Fast planning only vs. FaSTA\*.** To demonstrate the importance of the low-level fallback ("slow planning"), we evaluated a variant that uses **only** the high-level subroutines for subtasks where

subroutines are possible. If a subroutine failed its VLM check or if no subroutine was selected ('None'), the planner simply failed for that subtask, without activating the low-level graph.

As shown in Table 5, relying solely on the high-level subroutines leads to a significant drop in quality compared to our full adaptive fast-slow planning approach. While the cost is slightly lower (due to avoiding any low-level search), the brittleness is unacceptable. Figure 6 illustrates a typical failure case. While the High-Level Only approach fails the task, FaSTA* successfully recovers by falling back to the low-level A* search, demonstrating the critical role of the "slow planning" component for robustness and overall task success.

**Slow planning only vs FaSTA\*.** This ablation represents the original CoSTA* algorithm (using the refined single-path prompt). As shown in Table 5, CoSTA* achieves marginally higher output quality than FaSTA*. However, this slight quality gain comes at the expense of significantly higher average execution costs. The broader performance comparison, detailed in Table 3 and Figures 3 and 5, further highlights our superior cost-efficiency.

**Impact of LLM Size.** We evaluated a smaller model (Qwen 2.5VL 3B) for subroutine selection on a subset of tasks. The model selected incorrect subroutines $\sim$40% of the time, frequently triggering the fallback to the low-level A* search. Consequently, despite negligible API costs, the total execution cost increased by $\sim$20% due to the additional cost incurred by the fallback mechanism. However, the final output quality remained consistent, confirming the robustness of the slow-planning safety net against planner errors.

### 5.4 SUMMARY AND LIMITATIONS

Our results indicate that FaSTA*'s adaptive fast-slow execution strategy significantly improves computational efficiency while largely maintaining the high quality of the original CoSTA* approach. Ablation studies confirm the necessity of both the low-level fallback mechanism and the subroutine verification process for robustness and efficiency. Despite its promising performance, FaSTA* has a few limitations. Its initial performance on entirely new tasks may be suboptimal until sufficient experiences are gathered for effective subroutine learning (cold start). These limitations and a quantitative analysis of failure cases have been provided in Appendix C. Furthermore, overall performance and the ability to capture nuanced rules for complex tasks remain dependent on the evolving capabilities of the underlying LLMs.

## 6 CONCLUSION

In this paper, we introduced FaSTA*, a neurosymbolic agent designed to significantly enhance the efficiency of complex, multi-turn image editing tasks. By integrating an online subroutine mining mechanism that leverages LLM-based inductive reasoning in a modified version of In-Context Reinforcement Learning, FaSTA* learns and refines a library of effective tool-use sequences and their activation conditions from experiences. This learned knowledge underpins an adaptive fast-slow execution strategy, where a "Fast Plan" of subroutines made by LLMs is prioritized, with a localized A* search ("Slow Plan") serving as a robust fallback for novel or challenging subtasks. Our experiments demonstrate that FaSTA* achieves image quality comparable to the state-of-the-art CoSTA* but at a substantially reduced computational cost, effectively addressing a key bottleneck in prior methods. We believe that FaSTA*'s approach of combining learned symbolic shortcuts with principled search offers a promising direction for developing more agile, cost-sensitive, and continually improving AI agents for complex tasks.

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

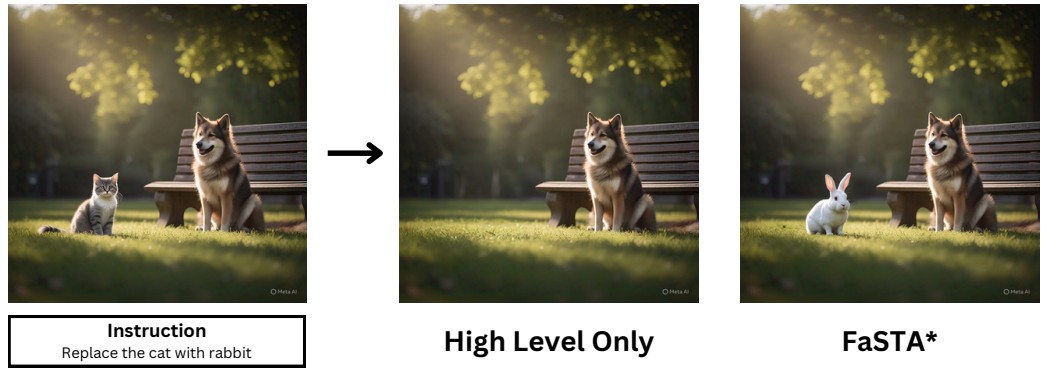

Figure 6: Failure case for "High-Level Only" execution versus FaSTA$^*$. For the task "Replace the cat with rabbit", the initially selected high-level subroutine fails to produce a satisfactory result, leading to a failed output for the "High-Level Only" approach. In contrast, FaSTA$^*$ detects this failure, activates its low-level fallback mechanism for the "Object Replacement" subtask, and performs A$^*$ search to find a correct tool sequence, successfully completing the task.

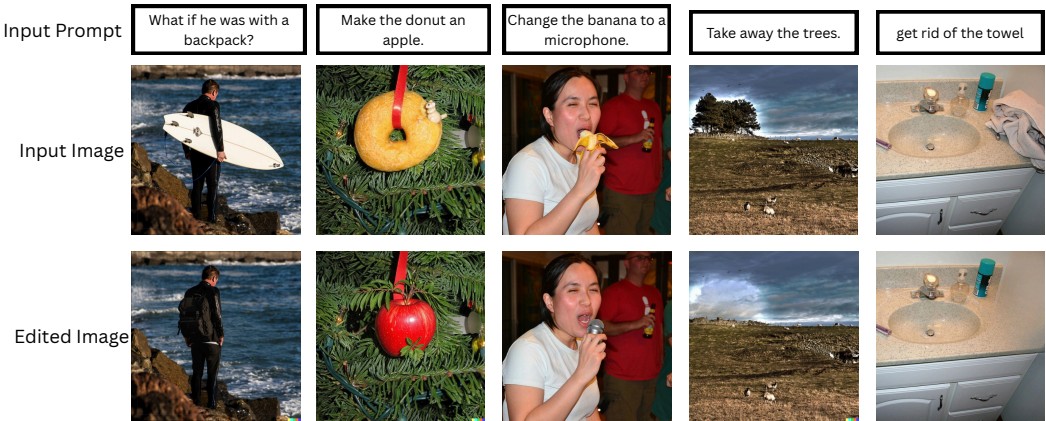

Figure 7: Qualitative examples of FaSTA$^*$'s performance on sample tasks from the MagicBrush dataset (Zhang et al., 2024a). These tasks were processed using the Subroutine Rule Table learned from non-benchmark data. Notably, all examples shown were successfully completed by FaSTA$^*$ relying entirely on its "fast plan" composed of learned subroutines, without needing to resort to the "slow planning" via A* search for any subtask.

## A   GENERALIZATION TO EXTERNAL DATASETS

### A.1   QUALITATIVE GENERALIZATION TO MAGICBRUSH DATASET

To assess the generalizability of FaSTA$^*$'s learned subroutines and its adaptive planning strategy, we tested it on a set of sample tasks derived from a different benchmark, the MagicBrush dataset (Zhang et al., 2024a). The Subroutine Rule Table ($\mathcal{R}$) used for these evaluations was the one learned through the online inductive reasoning process described in Section 4.2, which utilized diverse non-benchmark image/prompt pairs, ensuring no direct exposure to MagicBrush data during the subroutine learning phase.

Our objective here was to observe if the subroutines and the Fast Plan generation logic could effectively handle tasks from a dataset with potentially different characteristics and prompt styles without requiring immediate fallback to low-level A* search for every step. Figure 7 presents qualitative results for several such tasks.

As illustrated in Figure 7 and noted in its caption, FaSTA* was able to successfully complete these diverse editing tasks from the MagicBrush dataset. Significantly, for all the examples shown, the tasks were accomplished entirely through the "Fast Plan" execution. This indicates that the learned subroutines and their activation rules, derived from different data sources, were sufficiently general to apply effectively to these new instances, and the LLM was able to compose a successful Fast Plan without needing to trigger the low-level A* search fallback for any subtask. This provides positive evidence towards the generalizability of our subroutine learning and adaptive planning approach beyond the specific characteristics of the data used during the online refinement cycles.

## B BENCHMARK DATASET RATIONALE: CoSTA* VS. MAGICBRUSH

For evaluating FaSTA*, we primarily utilized the benchmark dataset introduced with CoSTA* (Gupta et al., 2025). While the MagicBrush dataset (Zhang et al., 2024a) is a prominent benchmark for instruction-guided image editing, the CoSTA* dataset was chosen due to its specific characteristics that better align with the capabilities we aim to demonstrate with FaSTA*, particularly its efficiency in handling complex, multi-step tasks.

The key distinctions motivating our choice are:

- **Task Complexity and Depth:** To best showcase FaSTA*'s advantages in cost-saving through learned subroutines and adaptive planning, complex examples involving a greater number of sequential subtasks are essential. The CoSTA* dataset was specifically curated to include such multi-turn editing instructions. In contrast, many tasks in the MagicBrush benchmark, while diverse, often involve simpler, more direct edits that might not fully stress or benefit from a sophisticated fast-slow planning approach to the same extent.

- **Multimodal Capabilities:** A significant aspect of modern image editing involves text-in-image manipulation. The CoSTA* dataset includes a substantial portion of tasks requiring multimodal processing (e.g., text replacement, text removal within an image context). The MagicBrush benchmark, as per its original release, primarily focuses on visual edits and does not extensively cover these text-in-image editing scenarios. FaSTA*, like CoSTA, is designed to handle such multimodal tasks, making the CoSTA* dataset more suitable for a comprehensive evaluation of its capabilities.

- **Number of Manipulations/Turns:** While the CoSTA* dataset has 121 image-prompt pairs, which might be fewer than the test set size of some other benchmarks like MagicBrush, the tasks in the CoSTA* dataset are designed to be multi-turn. As noted in Section 5, these tasks involve 1-8 subtasks per image, amounting to 550 total distinct image manipulations (or "turns") across the dataset. This provides a rich set of complex sequences for evaluating planning efficiency.

Table 6: Conceptual Comparison of Dataset Characteristics Relevant to FaSTA*.

| Characteristic | CoSTA* Dataset | MagicBrush Dataset |
|---|---|---|
| Primary Focus | Complex, Multi-Turn, Multimodal | Instruction-Guided Visual Edits |
| Max Subtasks per Task | 8 | 3 |
| Max Tool Subgraph Depth | 22 | 7 |
| Text-in-Image Editing | Supported | Not supported |

Table 6 provides a conceptual comparison of key characteristics relevant to our evaluation goals. While MagicBrush is invaluable for evaluating general instruction-following in image editing models, the CoSTA* dataset's emphasis on longer, multi-faceted tasks involving a broader range of subtask types (including multimodal ones) makes it a more fitting benchmark to demonstrate the specific cost-saving and adaptive planning strengths of FaSTA*. The goal of FaSTA* is not just to perform an edit, but to do so efficiently by learning and reusing common multi-step patterns, a capability best tested by complex sequential decision-making problems.

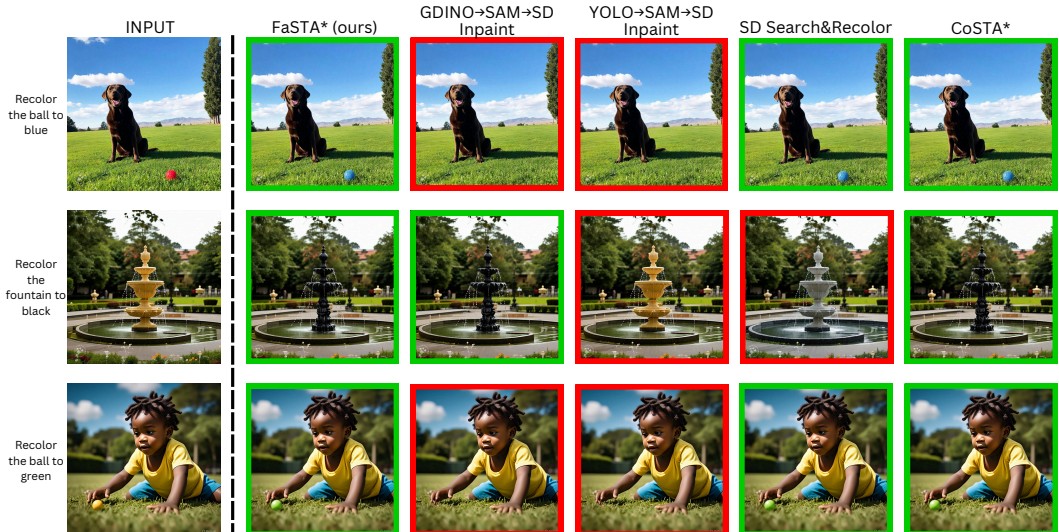

Figure 8: Example demonstrating FaSTA*'s subroutine effectiveness. FaSTA* uses learned rules to select optimal paths (e.g., `SD Search&Recolor` for the small ball in row 1, avoiding `SD Inpaint`'s potential failure), achieving results identical to CoSTA* at significantly lower average cost (15.21s vs. 25.32s for these examples) by preventing unnecessary exploration of suboptimal paths.

## C    DETAILED QUALITATIVE ANALYSIS AND SUBROUTINE EFFECTIVENESS

This section provides a more in-depth qualitative examination of FaSTA*'s performance, focusing on the benefits of learned subroutines and the efficiency gains of the adaptive fast-slow execution strategy.

**Demonstrating Subroutine Relevance and Efficiency.**    To illustrate why learning subroutines is beneficial and how FaSTA* achieves cost savings, we examined specific cases where input image and prompt conditions matched learned activation rules $\mathcal{C}$ for particular subroutines $\mathcal{P}$. Figure 8 depicts such scenarios for recoloring tasks, comparing the outcomes and behavior of FaSTA*, CoSTA*, and all potential tool paths for recoloration. Under specific conditions, certain tool paths are prone to failure or are suboptimal. FaSTA*, by leveraging its learned activation rules, can preemptively select an effective subroutine, thus avoiding unnecessary exploration of these less suitable paths.

For instance, consider the first row in Figure 8, where the task is to recolor a small ball to blue. Paths involving `SD Inpaint` (such as G-DINO→SAM→SD Inpaint (Rombach et al., 2022)or YOLO→SAM→SD Inpaint) often exhibit poor performance or fail VLM quality checks when the target object is very small. FaSTA* incorporates an activation rule for its chosen subroutine (in this case, one that selects `SD Search&Recolor`) that accounts for this factor. Thus, FaSTA* directly employs `SD Search&Recolor` and successfully recolors the ball. In contrast, CoSTA*, lacking this specific learned rule for this context, might explore an `SD Inpaint`-based path first due to its general applicability or heuristic score. Upon failure of this path (indicated by a failed VLM check), CoSTA* would then use its A* search to backtrack and explore alternative paths, eventually finding the `SD Search&Recolor` sequence and completing the task, but at the cost of the initial failed attempt and additional search time.

This pattern, where FaSTA* makes a more informed initial path choice due to its learned subroutine rules, is key to its efficiency. The figure shows that while both FaSTA* and CoSTA* can arrive at the same high-quality final output, FaSTA* does so more directly. For the examples presented in Figure 8, this translated to FaSTA* achieving the correct edits at an average execution cost of 15.21s, whereas CoSTA* took an average of 25.32s. This highlights how FaSTA*, by encoding successful, context-dependent tool sequences as subroutines and applying them based on learned activation

rules, effectively halves the execution cost in these scenarios by minimizing costly trial-and-error exploration of paths known to be suboptimal or likely to fail under specific conditions.

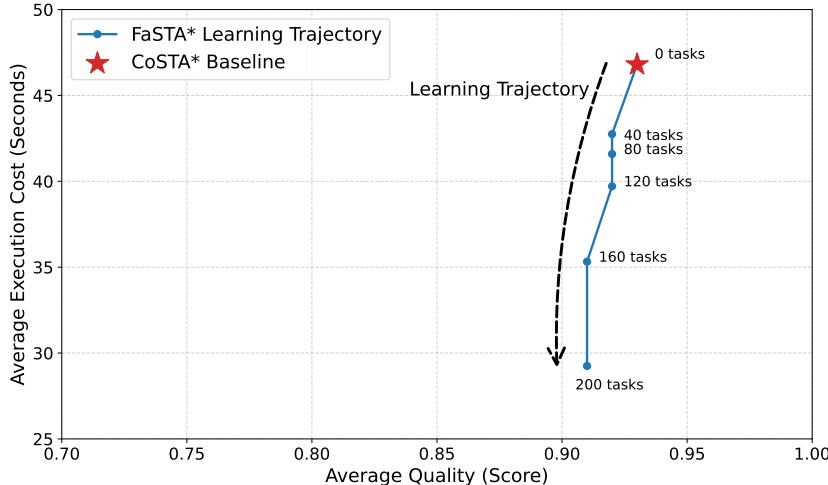

Figure 9: Learning trajectory of FaSTA* showing the relationship between Average Execution Cost and Average Quality as more tasks are explored (from 0 to 200). FaSTA* rapidly converges toward the Pareto frontier, significantly reducing cost while maintaining high quality.

### C.1    COST/QUALITY TRADE-OFF W.R.T. SUBROUTINE MINING

FaSTA* is designed to get optimal in terms of execution cost as more tasks are executed and subroutines are learnt. There is however, no degradation of final quality in any case. At "cold start" (0 explored tasks), FaSTA*'s cost and quality are identical to the CoSTA* baseline. FaSTA* begins to outperform FaSTA* in average efficiency as soon as the first effective subroutine is learned and successfully applied. The table below illustrates how performance evolves, showing a rapid decrease in cost while quality remains high and stable. This learning dynamics is visualized in Figure 9, demonstrating how the agent converges toward the Pareto frontier as it gains experience.

Table 7: Performance evolution of FaSTA*. Execution cost rapidly decreases as more subroutines are learned from explored tasks, while maintaining high quality comparable to the CoSTA* baseline.

| No. of Explored Tasks | Avg. Quality (FaSTA*) | Avg. Exec. Cost (s) (FaSTA*) | Avg. Quality (CoSTA* w/ Subtask Chain) (const.) | Avg. Exec. Cost (s) (CoSTA* w/ Subtask Chain) (const.) |
|---|---|---|---|---|
| 0 (Cold Start) | 0.93 | 46.8 | 0.93 | 46.8 |
| 40 | 0.92 | 42.75 | 0.93 | 46.8 |
| 80 | 0.92 | 41.59 | 0.93 | 46.8 |
| 120 | 0.92 | 39.71 | 0.93 | 46.8 |
| 160 | 0.91 | 35.32 | 0.93 | 46.8 |
| 200 (Warmed Up) | 0.91 | 29.25 | 0.93 | 46.8 |

We recognize that while FaSTA* covers most editing tasks, there might be novel OOD test cases where no subroutine applies. In such cases, FaSTA* gracefully defaults to its "slow path". This makes its performance for that specific task identical to CoSTA*'s, ensuring that output quality is not compromised, while the overall execution cost across all tasks is either identical or significantly better.

**Total Cost Analysis including LLM Overhead.** We further analyze the total inference cost to confirm that our reported efficiency gains hold even when accounting for LLM API overheads. Our comparisons with baselines are conducted under identical conditions, including and excluding specific factors uniformly. The primary computational bottleneck in agentic image editing is vision model inference (e.g., Stable Diffusion), which is orders of magnitude more expensive than text processing. By avoiding 10–20 exploratory image generation steps via Fast Planning, FaSTA* achieves net savings that vastly outweigh the cost of LLM tokens. The only unique overhead in our framework is the Inductive Reasoning phase; however, this cost is amortized over the agent's lifespan. As the system reaches optimal performance after ∼200 tasks (the "warm-up" phase), the reasoning overhead becomes negligible in long-term deployment. For instance, when amortized over just 2,000 tasks, the average cost reduction adjusts only slightly from 49.3% to ∼45.2%, maintaining a substantial efficiency advantage over baselines.

## C.2 FASTA* FAILURE CASE ANALYSIS

Overall, our adaptive "fast plan" is highly successful, handling 91% of subtasks efficiently. The more robust but costly "slow path" (A* search) is only triggered in the remaining 9% of cases. This fallback rate can be broken down as follows:

Table 8: Fallback reasons and their frequency.

| Fallback Reason | Percentage of Subtasks |
|---|---|
| Selected Subroutine Failed VLM Quality Check | 7% |
| No Applicable Subroutine Found | 2% |

The 7% of subroutine failures typically occur on novel tasks where the image context satisfies a subroutine's learned activation rules but contains unforeseen complexities, leading to a VLM quality check failure. The contribution to the 9% fallback rate varies by the complexity of the subtask category, as detailed below:

Table 9: Per-category contribution to fallback rate.

| Subtask Category | Contribution to Total Fallback Rate |
|---|---|
| Text Removal | 2.5% |
| Object Removal | 2.0% |
| Text Replacement | 1.8% |
| Object Replacement | 1.5% |
| Object Recoloration | 1.2% |

Even in these fallback scenarios, the system defaults to the robust A* search, ensuring that the final output quality is not compromised.

Table 10: Fair comparison of FaSTA* with VisProg and CLOVA using a restricted toolset and subtask list. Performance difference (Diff. w/ FaSTA*) is calculated as the average of baselines vs. FaSTA*.

| Subtasks | CoSTA* | FaSTA* | VisProg | CLOVA | Diff. w/ FaSTA* |
|---|---|---|---|---|---|
| 1-2 Subtasks | 0.510 | 0.509 | 0.498 | 0.504 | -1.6% |
| 3-4 Subtasks | 0.551 | 0.549 | 0.489 | 0.496 | -10.3% |
| 5-6 Subtasks | 0.573 | 0.571 | 0.446 | 0.451 | -21.4% |
| 7-8 Subtasks | 0.460 | 0.456 | 0.301 | 0.310 | -33.0% |
| **Overall Accuracy** | 0.525 | 0.521 | 0.436 | 0.446 | **-15.4%** |

Table 11: Fair comparison of FaSTA* with GenArtist using a restricted toolset and subtask list. Performance difference (Diff. w/ FaSTA*) is calculated as GenArtist vs. FaSTA*.

| Subtasks | CoSTA* | FaSTA* | GenArtist | Diff. w/ FaSTA* |
|---|---|---|---|---|
| 1-2 Subtasks | 0.508 | 0.507 | 0.506 | -0.2% |
| 3-4 Subtasks | 0.578 | 0.575 | 0.548 | -4.7% |
| 5-6 Subtasks | 0.606 | 0.602 | 0.503 | -16.4% |
| 7-8 Subtasks | 0.515 | 0.505 | 0.392 | -22.4% |
| **Overall Accuracy** | 0.553 | 0.547 | 0.495 | **-9.5%** |

## D  FAIR COMPARISON WITH RESTRICTED TOOLSET AND SUBTASKS

To strictly address concerns regarding equitable comparison due to toolset differences, we performed additional ablation studies where FaSTA* and CoSTA* were restricted to the exact same toolsets and subtasks available to the baselines. This ensures that any observed performance difference is primarily due to the planning and execution strategy (Fast-Slow architecture and Inductive Learning) rather than the breadth of available tools.

As shown in Table 10 and Table 11, even when handicapped with the restricted toolset, FaSTA* achieves performance comparable to CoSTA* and significantly outperforms the baselines. This confirms that the performance gains stem from our architectural contributions rather than simply having access to more powerful tools.

## E  DETAILED OVERVIEW OF COSTA* COMPONENTS AND PLANNING

This appendix provides a more detailed explanation of the core components and planning stages of the CoSTA* agent (Gupta et al., 2025), which form the foundation upon which FaSTA* builds and modifies. The descriptions here are rephrased from the original CoSTA* paper to ensure clarity and avoid direct repetition.

### E.1  FOUNDATIONAL KNOWLEDGE STRUCTURES IN COSTA*

CoSTA* leverages three primary pre-defined knowledge structures to inform its planning and search processes:

**Model Description Table (MDT)**   The MDT serves as a comprehensive catalog of all AI tools available to the agent. For each tool (e.g., YOLO (Wang et al., 2022), SAM (Kirillov et al., 2023), Stable Diffusion (Rombach et al., 2022)), the MDT specifies:

- The types of subtasks it can perform (e.g., "Object Detection", "Image Segmentation", "Object Recoloration"). CoSTA* considered 24 tools supporting 24 distinct subtasks.
- Its input requirements (e.g., an image, bounding boxes, segmentation masks).
- The outputs it produces (e.g., bounding boxes, edited image, extracted text).

This structured information is essential for mapping abstract subtasks to concrete tool invocations and for constructing the Tool Dependency Graph. An excerpt of CoSTA*'s MDT structure is shown in (Gupta et al., 2025, Table 1), with the full table in their appendix.

**Tool Dependency Graph (TDG)**   The TDG, denoted $G_{td} = (V_{td}, E_{td})$, is a directed graph that captures the operational dependencies between the AI tools listed in the MDT.

- $V_{td}$ is the set of all available AI tools.
- An edge $(v_1, v_2) \in E_{td}$ exists if the output of tool $v_1$ can serve as a valid input for tool $v_2$ in the context of certain subtasks.

The TDG represents the potential workflow sequences. CoSTA$^*$ can automatically construct this graph by analyzing the input/output specifications of tools in the MDT, reducing manual effort and facilitating updates as new tools are added (Gupta et al., 2025, Appendix C). A visualization of the TDG used in CoSTA$^*$ is provided in (Gupta et al., 2025, Fig. 4).

**Benchmark Table (BT)**   The BT is a critical resource for the A$^*$ search's heuristic function. It stores pre-computed or empirically measured performance data for tool-subtask pairs $(v_i, s_j)$. For each pair, the BT typically includes:

- **Execution Time** $C(v_i, s_j)$**:** The average time taken by tool $v_i$ to perform subtask $s_j$.
- **Quality Score** $Q(v_i, s_j)$**:** An objective measure of the typical output quality of tool $v_i$ for subtask $s_j$, normalized to a [0,1] scale for comparability across different subtasks.

This data is sourced from existing benchmarks where available, or through dedicated offline evaluations if necessary, as detailed in (Gupta et al., 2025, Sec 3.3). The complete BT for CoSTA$^*$ is shown in (Gupta et al., 2025, Table 11).

### E.2   CoSTA$^*$ Planning and Execution Stages

The CoSTA$^*$ agent follows a three-stage process to address a given multi-turn image editing task:

**1. Task Decomposition and Subtask-Tree Generation**   Given an input image $x$ and a natural language instruction $u$, CoSTA$^*$ employs an LLM to decompose the complex request into a sequence of more manageable subtasks. This decomposition results in a *subtask tree*, $G_{ss} = (V_{ss}, E_{ss})$.

- Each node $v_i \in V_{ss}$ corresponds to a specific subtask $s_i$ (e.g., "remove car," "recolor bench to pink").
- Edges $(v_i, v_j) \in E_{ss}$ represent dependencies, indicating that subtask $s_i$ must be completed before $s_j$.
- The LLM uses a prompt template $f_{\text{plan}}(x, u, S)$ which includes the image, instruction, and a list of supported subtasks $S$. The original CoSTA$^*$ framework allowed for the generation of trees with multiple parallel paths if subtasks were independent, representing different valid execution orders.

This subtask tree provides a high-level plan for addressing the user's request.

**2. Tool Subgraph Construction**   The abstract subtask tree $G_{ss}$ is then translated into a concrete *Tool Subgraph* $G_{ts} = (V_{ts}, E_{ts})$, which is the actual graph the A$^*$ search will operate on.

- For each subtask node $s_i$ in $G_{ss}$, the MDT is consulted to find all tools $M(s_i)$ capable of performing $s_i$.
- The TDG is then used to backtrack from these tools to include all necessary prerequisite tools and their interconnections, forming a minimal tool subgraph $G_{td}^i$ for that subtask.
- The final $G_{ts}$ is formed by taking the union of all such $G_{td}^i$ and connecting them according to the dependencies specified in $G_{ss}$. This subgraph contains all feasible tool sequences for the given task.

This construction prunes the global tool graph to a smaller, task-relevant search space.

**3. Cost-Sensitive A$^*$ Search for Optimal Toolpath**   CoSTA$^*$ employs an A$^*$ search algorithm on the Tool Subgraph $G_{ts}$ to find an optimal toolpath that balances execution cost and output quality, according to a user-defined trade-off parameter $\alpha$.

- **Priority Function:** The A$^*$ search prioritizes nodes (representing tool executions) based on the function $f(x) = g(x) + h(x)$.
- **Actual Execution Cost** $g(x)$**:** This term represents the cumulative cost-quality score of the path taken so far to reach node $x$. It is computed in real-time as the execution progresses. For a path of $k$ tool-subtask executions $(v_j, s_j)$, $g(x)$ is defined as:

$$g(x) = \left( \sum_{j=1}^{x} c(v_j, s_j) \right)^{\alpha} \times \left( 2 - \prod_{j=1}^{x} q(v_j, s_j) \right)^{2-\alpha}$$

where $c(v_j, s_j)$ is the actual measured execution time of tool $v_j$ for subtask $s_j$, and $q(v_j, s_j)$ is the real-time quality score of its output, validated by a VLM. The parameter $\alpha$ controls the emphasis on cost versus quality (higher $\alpha$ prioritizes lower cost). These values include only the nodes in the current path and each node is initialized with $g(x) = \infty$ and updated if a lower values is found. The start node is initialized with $g(x) = 0$.

- **Heuristic Cost $h(x)$:** This term is an admissible estimate of the cost-to-go from the current node $x$ (representing a tool-task pair $(v_i, s_i)$) to a goal/leaf node in $G_{ts}$. It is pre-calculated using values from the Benchmark Table (BT) and considers the trade-off parameter $\alpha$. For a node $x$, $h(x)$ is recursively defined based on its neighbors $y$:

$$h(x) = \min_{y \in \text{Neighbors}(x)} \left( (h_C(y) + C(y))^\alpha \times (2 - Q(y) \times h_Q(y))^{2-\alpha} \right)$$

where $h_C(y)$ and $h_Q(y)$ are the cost and quality components of the heuristic for neighbor $y$ (initialized to 0 and 1 respectively for leaf nodes), and $C(y)$ and $Q(y)$ are the benchmark time and quality for tool $y$.

- **VLM Feedback and Retries:** After each tool execution, its output is evaluated by a VLM. If the quality score falls below a predefined threshold, CoSTA* can trigger a retry mechanism (e.g., by adjusting the tool's hyperparameters) and updates $g(x)$ with any additional costs. If retries fail, the A* search naturally explores alternative paths with better $f(x)$ scores, allowing robust recovery from tool mispredictions or failures.

This A* search process aims to find a toolpath that optimally satisfies the user's preference for cost versus quality.

## F   RATIONALE FOR USING SUBTASK CHAIN GENERATION

The original CoSTA* agent (Gupta et al., 2025) utilized an LLM prompt that could generate a *subtask tree* $G_{ss}$. This tree structure allowed for multiple parallel branches, representing alternative valid execution orders for subtasks that were independent of each other. While flexible, exploring these multiple branches during the subsequent A* search phase could potentially increase computational cost, especially for tasks with many independent subtasks.

We hypothesized that current Large Language Models (LLMs) and Vision-Language Models (VLMs) might possess improved reasoning capabilities to determine a single, logical sequence of subtasks based on the user prompt and image context directly. This improved reasoning could potentially determine a single, optimal, or at least highly plausible, logical sequence for the required subtasks upfront. Such a linear sequence, which we term a "subtask chain," would simplify the initial planning structure compared to a branching tree.

To validate the feasibility of using a simpler chain structure without sacrificing outcome quality, we conducted a preliminary experiment. We compared the performance of the original CoSTA* algorithm using two different prompts: the original prompt generating multi-path subtask trees, and a modified prompt designed to generate only a single-path subtask chain. We ran both versions on a representative subset of 50 diverse tasks from the benchmark dataset (Gupta et al., 2025), covering a range of subtask counts and types. The final output quality, assessed using human evaluation metrics defined in (Gupta et al., 2025), was found to be consistently very similar between the two approaches, with scores varying by only **1.08%** on average across the tested examples.

Given this negligible impact on final quality, we adopted the simpler single-path prompt to generate a subtask chain for FaSTA*, aiming to potentially reduce planning complexity. The specific prompt used for subtask chain generation in FaSTA* can be found in Appendix Q.

Further analysis focused on quantifying the cost impact of this prompt change within the CoSTA* framework itself, independent of FaSTA*'s subroutine mechanism. We compare the average execution cost of CoSTA* using the old (multi-path tree) prompt versus the new (single-path chain) prompt across full dataset.

Table 12 presents these cost results. Using the single-path (chain) prompt yielded a noticeable reduction in CoSTA*'s average execution time (from 58.2 to 46.8 seconds). This confirms that leveraging the LLM to generate a more constrained initial plan structure offers some efficiency benefits even for the original A* search approach.

Table 12: Impact of Subtask Tree Prompt on CoSTA*.

| CoSTA* Variant | Avg. Cost (s) |
|---|---|
| CoSTA* (Old Prompt - Multi-Path) | 58.2 |
| CoSTA* (New Prompt - Single Path) | 46.8 |

However, it is crucial to contextualize these savings. While the refined prompt contributes to efficiency, the primary driver of the substantial cost reduction observed in the full FaSTA* system (29.5s) compared to the baseline CoSTA* (even with the new prompt, 46.8s) is the adaptive fast-slow execution strategy (Section 4.3) that effectively utilizes learned subroutines.

## G  DETAILED ONLINE SUBROUTINE INDUCTION AND REFINEMENT PROCESS

This appendix provides a detailed technical breakdown of the online adaptation and refinement loop used by FaSTA* to learn and update its Subroutine Rule Table ($\mathcal{R}$), as introduced in Section 4.2.

### G.1  STEP 1: DATA LOGGING

For every subtask execution where multiple paths/tools possible, FaSTA* logs a comprehensive trace $\tau$. Each trace is structured to capture critical information necessary for subsequent learning. The components of a trace typically include:

- **Subtask Identification** ($s_j$): The specific type of subtask being addressed (e.g., 'Object Recoloration', 'Text Removal').
- **Final Executed Tool Path** ($\mathcal{P}_j$): The actual sequence of tools $(t_1, t_2, \ldots, t_k)$ that was executed by the planner (either a pre-existing subroutine or a path found via A* search during a "slow planning" phase) to complete $s_j$.
- **Context Features:** A rich set of features describing the state of the image and relevant objects at the time of executing $s_j$. These are gathered from various sources:
    - Outputs from perception tools like YOLO (Wang et al., 2022) (e.g., 'object_size' and SAM (Kirillov et al., 2023) (e.g., 'mask_properties', 'color_details').
    - Higher-level semantic features inferred by an LLM query at the start of the task for relevant objects (e.g., 'background_content_type', 'overlapping_critical_elements', 'object_clarity').

    (Further details on trace composition are in Appendix M).
- **Aggregated Path Cost** ($C_{path}$): The total execution cost (e.g., sum of tool runtimes) for the path $\mathcal{P}_j$.
- **Aggregated Path Quality** ($Q_{path}$): The overall quality of the outcome from $\mathcal{P}_j$, typically an average of VLM scores for the constituent tools.
- **Failure Information:** Details of any intermediate tool failures or VLM quality check failures encountered during the execution of $\mathcal{P}_j$, including the specific context features present at the moment of failure.

These traces are stored in a buffer $\mathcal{B}$.

### G.2  STEP 2: PERIODIC REFINEMENT TRIGGER

The learning process is not continuous but triggered periodically to manage computational load. After a predefined number of task inferences, $K$ (e.g., $K = 20$), the system initiates a refinement cycle. The $K$ most recent traces, $\mathcal{T}_{recent} = \{\tau_k\}_{k=t-K+1}^{t}$, are retrieved from the buffer $\mathcal{B}$ to serve as the input for the learning phase.

### G.3  STEP 3: INDUCTIVE REASONING BY LLM

The core of the learning process involves an LLM (OpenAI o1) performing inductive reasoning. The LLM is prompted with:

- The set of recent traces, $\mathcal{T}_{recent}$, which provide examples of successful and failed tool path executions under various contexts.
- The current Subroutine Rule Table, $\mathcal{R}$.

The LLM's task (guided by the prompt in Appendix S) is to analyze these inputs to identify patterns. It looks for correlations between context features, specific tool sequences (potential subroutines), and their observed execution outcomes (cost, quality, success/failure). Based on these identified patterns, the LLM proposes a set of potential changes, $\Delta_{proposals} = \{\Delta_1, \Delta_2, \dots\}$, to the rule set $\mathcal{R}$. Each proposed change $\Delta$ can be one of the following:

- Adding a new subroutine $\mathcal{P}_j$ along with its inferred activation rule $\mathcal{C}_j$ for a specific subtask $s_j$.
- Modifying the tool sequence $\mathcal{P}_j$ of an existing subroutine.
- Modifying the activation conditions $\mathcal{C}_j$ of an existing subroutine.

### G.4 STEP 4: VERIFICATION AND SELECTION OF NEW/MODIFIED SUBROUTINES

Each proposed change $\Delta \in \Delta_{proposals}$ undergoes a rigorous verification process before being accepted into the active Subroutine Rule Table $\mathcal{R}$. This ensures that only genuinely beneficial and robust rules are adopted. The verification for a single proposed change $\Delta$ (related to a subtask $s_\Delta$) proceeds as follows:

- **Subtask-Specific Test Datasets:** To evaluate $\Delta$, a specialized test dataset $\mathcal{D}_{s_\Delta}$ is used. This dataset, constructed from the CoSTA* benchmark images (Gupta et al., 2025), contains diverse tasks where all constituent subtask instances are exclusively of type $s_\Delta$. (Details in Appendix L).
- **Baseline Performance Establishment:** A baseline performance pair $(C_{base}, Q_{base})$ is determined by executing a baseline system on a randomly sampled test set $\mathcal{T}' \subset \mathcal{D}_{s_\Delta}$.
  - If this is the first refinement cycle (i.e., $t = K$), the baseline system is CoSTA*.
  - For subsequent cycles, the baseline is FaSTA* operating with its current, pre-change rule set $\mathcal{R}$.
- **Evaluation of Proposed Change:** The proposed change $\Delta$ is provisionally applied to the current rule set $\mathcal{R}$ to create a candidate rule set $\mathcal{R}'$. FaSTA* is then executed with $\mathcal{R}'$ on the *same* sampled test set $\mathcal{T}'$ to obtain new performance metrics $(C_{new}, Q_{new})$.
- **Performance Metrics Calculation:** The percentage changes in cost and quality are computed:

$$\Delta C\% = \frac{C_{new} - C_{base}}{C_{base}} \times 100$$

$$\Delta Q\% = \frac{Q_{new} - Q_{base}}{Q_{base}} \times 100$$

- **Acceptance Criterion (Net Benefit):** A Net Benefit score $B(\Delta)$ is calculated to quantify the overall impact of the change:

$$B(\Delta) = \Delta C\% - \Delta Q\%$$

A change $\Delta$ is considered beneficial and is provisionally accepted if $B(\Delta) < 0$. This criterion prioritizes changes that yield a greater percentage improvement in cost than any percentage degradation in quality, or improve quality with no cost increase, etc.
- **Retry Mechanism for Refinement:** If the initial proposed change $\Delta$ does not meet the Net Benefit criterion ($B(\Delta) \geq 0$), it is not immediately discarded. Instead, feedback detailing the failure (e.g., specific test cases where it underperformed, the nature of $C_{new}$ vs. $C_{base}$ and $Q_{new}$ vs. $Q_{base}$) is provided to the LLM. The LLM is then prompted to refine its initial proposal, yielding a modified change $\Delta'$. This refinement-evaluation cycle (using a *new* random test sample $\mathcal{T}'' \subset \mathcal{D}_{s_\Delta}$ for re-evaluation) can be repeated up to $N_{retries}$ times (e.g., $N_{retries} = 2$).
- **Final Decision:** The proposed change (either the original $\Delta$ or a refined $\Delta'$) is permanently accepted and integrated into the main Subroutine Rule Table $\mathcal{R}$ only if it satisfies the $B(\Delta) < 0$ criterion within the allowed number of retries. If, after all retries, the criterion is still not met, the proposed change is discarded for this refinement cycle.

This comprehensive verification loop ensures that the Subroutine Rule Table evolves with high-quality, empirically validated rules.

## H    EFFICACY OF ONLINE SUBROUTINE LEARNING

To demonstrate the progressive effectiveness of our online subroutine induction and refinement process (Section 4.2), we analyzed how the performance of FaSTA$^*$'s "Fast Plan" improved over time. This involved tracking the success rate of the high-level Fast Plan when applied to the main benchmark dataset at various stages of the learning process.

The learning itself was driven by execution traces from tasks *distinct* from this benchmark dataset. Specifically, the data fed to the LLM for inductive reasoning (at $K = 40, 80, 120, \ldots$ cumulative external task intervals) was generated from a continuously expanding set of diverse, newly created prompts or random samples from broader image collections. This separation of "training/learning" tasks from the "monitoring" benchmark tasks was crucial to ensure that the observed improvements in Fast Plan success were due to genuine generalization of learned subroutines and not overfitting to the benchmark data itself.

Figure 1 illustrates this learning efficacy. It shows the percentage of applicable subtasks within the held-out benchmark portion for which FaSTA$^*$ successfully utilized a learned subroutine (i.e., the Fast Plan step was not 'None' and did not require a fallback to the Slow Path due to VLM failure). This success rate is plotted against the cumulative number of non-benchmark "training" task executions that had been processed to refine the Subroutine Rule Table up to that point. The analysis focuses on subtasks where multiple tool paths are typically possible, making them prime candidates for subroutine mining. The bar chart displays the Fast Plan success rate at discrete intervals (e.g., after the Rule Table was updated based on 40, 80, 120, 160, and 200 external task evaluations), with an overlaid curve highlighting the improvement trend. The final Subroutine Rule Table used for the main benchmark evaluations reported in Section 5 is the one achieved after a substantial number of such online learning and refinement iterations.

The increasing trend observed in Figure 1 demonstrates that as FaSTA$^*$ processes more diverse external tasks and iteratively refines its Subroutine Rule Table, its ability to successfully apply efficient, learned "fast plans" to unseen benchmark tasks improves. This, in turn, reduces the frequency of needing to resort to the more computationally intensive "slow path" A$^*$ search for subtask types amenable to subroutine learning, contributing to the overall efficiency reported in our main results.

## I    COMPLETE SUBROUTINE RULE TABLE

The complete Subroutine Rule Table ($\mathcal{R}$) used by FaSTA$^*$ is detailed below. This table stores the learned subroutines ($\mathcal{P}_j$), their symbolic activation rules ($\mathcal{C}_j$) based on context features, the associated subtask ($s_j$), and the empirically measured average execution cost and quality observed during execution of evaluation tasks (Section 4.2). **It is important to note that the inductive reasoning process for mining subroutines focuses primarily on subtasks where multiple viable tool sequences or configurations exist (e.g., object replacement, object recoloration, text removal)**, offering potential for optimization via learned rules. Subtasks typically solved by a single, fixed tool path (e.g., depth estimation using MiDaS, basic text detection using CRAFT (Baek et al., 2019)) are generally not subjected to this mining process and thus may not appear with complex rules or multiple subroutine options in this table. It should also be noted while most information used in traces used for inductive reasoning is obtained as inputs from outputs of intermediate tools along the path (eg. Object Size from YOLO (Wang et al., 2022), etc.), some information like the `background_content_type`, etc. is obtained by prompting the LLM separately on the image. This table is dynamically updated during the online refinement process.

## J    COMPARISON WITH OTHER STRATEGIES

### J.1    WITH DIRECT CACHING

While LLM-based subroutine induction is the most suitable approach to learn symbolic rules in a generalizable manner, the high-level similarity with direct caching might raise the question as to which is this a better approach than the latter. A "direct cache" or memorization-based baseline is too brittle. It would require a new task to have nearly identical conditions to a previously seen one to be

Table 13: Complete Learned Subroutine Rule Table ($\mathcal{R}$). Contains all mined subroutines, activation rules, and performance metrics used in the Fast Plan generation.

| Subtask | Subroutine $\mathcal{P}$ | Name | Activation Rules $\mathcal{C}$ | Avg. Cost (s) | Avg. Quality |
|---|---|---|---|---|---|
| Object Recoloration | Grounding DINO (Liu et al., 2024) → SAM (Kirillov et al., 2023) → SD Inpaint (Rombach et al., 2022) | SR1 | - `object_size`: Not Too Small
- `overlapping_critical_elements`: None (eg. Some text written on object to be recolored and this text is critical for some future or past subtask) | 10.39 | 0.89 |
| Object Recoloration | SD Search & Recolor (Rombach et al., 2022) | SR2 | - `color_transition` = not extreme luminance change (e.g., not White ↔ Black) | 12.92 | 0.95 |
| Object Recoloration | YOLO (Wang et al., 2022) → SAM (Kirillov et al., 2023) → SD Inpaint (Rombach et al., 2022) | SR3 | - `yolo_class_support`: Object supported as a yolo class
- `object_size`: Not Too Small
- `overlapping_critical_elements`: None (eg. Some text written on object to be recolored and this text is critical for some future or past subtask) | 10.36 | 0.88 |
| Object Replacement | Grounding DINO (Liu et al., 2024) → SAM (Kirillov et al., 2023) → SD Inpaint (Rombach et al., 2022) | SR4 | - `object_size` = Not too small
- `size_difference`(original, target objects) = Not too big (eg. hen to car, etc)
- `shape_difference`(original, target objects) = Not too small (i.e., not confusingly similar, eg. bench and chair) | 10.41 | 0.91 |
| Object Replacement | SD Search&Replace (Rombach et al., 2022) | SR5 | - `instance_count`(object_to_replace) = 1, 2
- `object_clarity` = High (e.g., common, opaque, substantial, fully visible)
- `shape_difference`(original, target) = Not Very Large | 12.12 | 0.97 |
| Object Replacement | YOLO (Wang et al., 2022) → SAM (Kirillov et al., 2023) → SD Inpaint (Rombach et al., 2022) | SR6 | - `yolo_class_support`: Object supported as a yolo class
- `object_size` = Not too small
- `size_difference`(original, target objects) = Not too big (eg. hen to car, etc)
- `shape_difference`(original, target objects) = Not too small (i.e., not confusingly similar, eg. bench and chair) | 10.38 | 0.91 |
| Object Removal | Grounding DINO (Liu et al., 2024) → SAM (Kirillov et al., 2023) → SD Erase (Rombach et al., 2022) | SR7 | - `object_size` = Not too big
- `background_content_type` = Simple_Texture OR Homogenous_Area OR Repeating_Pattern (e.g., wall, sky, grass, water, simple ground)
- `background_reconstruction_need` = Filling/Inpainting (vs. Drawing/Semantic_Completion) | 11.97 | 0.98 |
| Object Removal | YOLO (Wang et al., 2022) → SAM (Kirillov et al., 2023) → SD Erase (Rombach et al., 2022) | SR8 | - `yolo_class_support`: Object supported as a yolo class
- `object_size` = Not too big
- `background_content_type` = Simple_Texture OR Homogenous_Area OR Repeating_Pattern (e.g., wall, sky, grass, water, simple ground)
- `background_reconstruction_need` = Filling/Inpainting (vs. Drawing/Semantic_Completion) | 11.95 | 0.98 |
| Object Removal | Grounding DINO (Liu et al., 2024) → SAM (Kirillov et al., 2023) → SD Inpaint (Rombach et al., 2022) | SR9 | - `object_size` = Not small
- `background_content_type` = Complex_Scene OR Occludes_Specific_Objects
- `background_reconstruction_need` = Drawing/Semantic_Completion (vs. Filling/Inpainting) | 10.39 | 0.95 |
| Object Removal | YOLO (Wang et al., 2022) → SAM (Kirillov et al., 2023) → SD Inpaint (Rombach et al., 2022) | SR10 | - `yolo_class_support`: Object supported as a yolo class
- `object_size` = Not small
- `background_content_type` = Complex_Scene OR Occludes_Specific_Objects
- `background_reconstruction_need` = Drawing/Semantic_Completion (vs. Filling/Inpainting) | 10.37 | 0.95 |
| Text Removal | CRAFT (Baek et al., 2019) → EasyOCR (JaidedAI, 2024)+DeepFont → LLM → SD Erase | SR11 | - `background_content_behind_text` = Plain_Color OR Simple_Gradient OR Simple_Texture (Not Complex_Image or Specific_Objects)
- `background_reconstruction_need` = Filling/Inpainting (vs. Drawing/Semantic_Completion) | 17.81 | 0.93 |
| Text Removal | CRAFT (Baek et al., 2019) → EasyOCR+DeepFont (Wang et al., 2015) → LLM → DALL-E | SR12 | - `background_artifact_tolerance` = High (e.g., clouds, noisy textures, abstract patterns where minor flaws are acceptable)
- `surrounding_context_similarity`(to_text) = Low (e.g., nearby areas do not contain other text or fine line patterns) | 17.95 | 0.96 |
| Text Removal | CRAFT (Baek et al., 2019) → EasyOCR+DeepFont → LLM → Painting | SR13 | - `background_content_behind_text` = Uniform_Solid_Color (Strictly no texture, gradient, or objects)
- `background_reconstruction_need` = None (Simple solid color fill is sufficient) | 6.69 | 0.95 |
| Text Replacement | CRAFT → EasyOCR+DeepFont → LLM → SD Erase → Text Writing | SR14 | - `background_content_behind_text` = Plain_Color OR Simple_Gradient OR Simple_Texture (Not Complex_Image or Specific_Objects)
- `background_reconstruction_need` = Filling/Inpainting (vs. Drawing/Semantic_Completion) | 17.85 | 0.92 |
| Text Replacement | CRAFT → EasyOCR+DeepFont → LLM → DALL-E → Text Writing | SR15 | - `background_artifact_tolerance` = High (e.g., clouds, noisy textures, abstract patterns where minor flaws are acceptable)
- `surrounding_context_similarity`(to_text) = Low (e.g., nearby areas do not contain other text or fine line patterns) | 18.02 | 0.94 |
| Text Replacement | CRAFT → EasyOCR+ DeepFont → LLM → Painting → Text Writing | SR16 | - `background_content_behind_text` = Uniform_Solid_Color (Strictly no texture, gradient, or objects)
- `background_reconstruction_need` = None (Simple solid color fill is sufficient) | 6.77 | 0.93 |

effective, which is rare. We ran a preliminary study on such a baseline, which reused a toolpath if a new task's context features (e.g., object size) were within a tight 5% tolerance of a cached example.

The results show that even with this tolerance, the fallback rate remains extremely high, as it struggles to generalize. The table below compares the fallback rate of this direct cache baseline against FaSTA*'s, clearly illustrating the significant benefit of our inductive reasoning approach.

This comparison demonstrates that FaSTA*'s ability to learn generalized, symbolic rules is far more effective than simple memorization, allowing it to successfully handle a much broader range of unseen tasks.

Table 14: Fallback rates vs. explored tasks.

| Explored Tasks | Direct Cache Fallback Rate (%) | FaSTA* Fallback Rate (%) |
|---|---|---|
| 40 | 98% | 79% |
| 80 | 98% | 73% |
| 120 | 97% | 62% |
| 160 | 96% | 43% |
| 200 | 95% | 9% |

## J.2 WITH END-TO-END MODELS

We compared FaSTA* with a state-of-the-art closed-source model, the Gemini 2.0 Flash Preview Image Generation, on a few tasks from our benchmark. The qualitative results, which highlight FaSTA*'s ability to adhere to complex instructions, are presented in Figure 13. We have also conducted further quantitative comparisons on our evaluation benchmark with both Gemini and GPT-4o. The results are summarized below:

Table 15: Comparison of Quality Scores with End-to-end Models

| Method | Average Quality Score |
|---|---|
| **Ours** | 0.91 |
| **Gemini 2.0** | 0.78 |
| **GPT-4o (with image editing)** | 0.74 |

## K    HUMAN EVALUATION METHODOLOGY FOR ACCURACY

To ensure a robust and reliable assessment of model performance, particularly for complex, multi-step, and multimodal editing tasks where automated metrics like CLIP (Radford et al., 2021) similarity can be insufficient (Gupta et al., 2025), we employ human evaluation to measure accuracy. Automated metrics often fail to capture nuanced errors, semantic inconsistencies, or critical local changes within complex edits (Gupta et al., 2025, Sec 5.2, Appx J). Our structured human evaluation process provides a more accurate measure of task success. The variance in scores among evaluators was 0.07, indicating strong inter-evaluator consistency.

### K.1 SUBTASK-LEVEL ACCURACY SCORING

Human evaluators manually assess the output of each individual subtask $s_i$ within a larger task $T$. Each subtask is assigned a correctness score, denoted as $A(s_i)$, based on the following scale:

- $A(s_i) = 1$, if the subtask is completed fully and correctly.
- $A(s_i) = 0$, if the subtask execution failed entirely or produced an unusable result.
- $A(s_i) = x$, where $x \in \{0.1, 0.3, 0.5, 0.7, 0.8, 0.9\}$, if the subtask is partially correct (Gupta et al., 2025, Eq. 4).

The specific score $x$ for partial correctness is determined using predefined rules tailored to different types of editing operations, ensuring consistency in evaluation. These rules, adapted from (Gupta et al., 2025), are outlined in Table 16.

### K.2 TASK-LEVEL ACCURACY CALCULATION

The accuracy for a complete task $T$, denoted as $A(T)$, is calculated as the arithmetic mean of the correctness scores of all its constituent subtasks $S_T$ (Gupta et al., 2025, Eq. 5):

$$A(T) = \frac{1}{|S_T|} \sum_{s_i \in S_T} A(s_i)$$

Table 16: Predefined Rules (adapted from (Gupta et al., 2025, Table 8)) for Assigning Partial Correctness Scores in Human Evaluation.

| Task Type | Evaluation Criteria | Assigned Score |
|---|---|---|
| Image-Only Tasks | Minor artifacts, barely noticeable distortions | 0.9 |
| | Some visible artifacts, but main content is unaffected | 0.8 |
| | Noticeable distortions, but retains basic correctness | 0.7 |
| | Significant artifacts or blending issues | 0.5 |
| | Major distortions or loss of key content | 0.3 |
| | Output is almost unusable, but some attempt is visible | 0.1 |
| Text+Image Tasks | Text is correctly placed but slightly misaligned | 0.9 |
| | Font or color inconsistencies, but legible | 0.8 |
| | Noticeable alignment or formatting issues | 0.7 |
| | Some missing or incorrect words but mostly readable | 0.5 |
| | Major formatting errors or loss of intended meaning | 0.3 |
| | Text placement is incorrect, missing, or unreadable | 0.1 |

This approach ensures that the task-level accuracy reflects the performance across all required steps.

### K.3 OVERALL SYSTEM ACCURACY

To evaluate the overall performance of the system across the entire benchmark dataset, the overall accuracy $A_{overall}$ is computed by averaging the task-level accuracies $A(T_j)$ for all evaluated tasks $T_j$ (Gupta et al., 2025, Eq. 6):

$$A_{overall} = \frac{1}{|T|} \sum_{j=1}^{|T|} A(T_j)$$

where $|T|$ represents the total number of tasks evaluated in the dataset.

## L SUBROUTINE VERIFICATION: DATASETS AND EVALUATION PROTOCOL

This section provides further details on the datasets and evaluation procedure used for verifying proposed subroutine changes ($\Delta$) during the online refinement process described in Section G.4.

### L.1 SUBTASK-SPECIFIC TEST DATASETS ($\mathcal{D}_{s_\Delta}$)

To rigorously evaluate a proposed change $\Delta$ (which typically relates to a specific subtask type $s_\Delta$, e.g., Object Replacement), we created specialized test datasets, one for each subtask type supported by the system ($\mathcal{D}_{s_\Delta}$).

- **Image Source:** These datasets reuse the base images from the original CoSTA* benchmark dataset (Gupta et al., 2025). Image-based subtask datasets (e.g., $\mathcal{D}_{\text{Object Replacement}}$, $\mathcal{D}_{\text{Object Recoloration}}$) utilize all 121 images. Text-related subtask datasets (e.g., $\mathcal{D}_{\text{Text Removal}}$) utilize the subset of 40 images from the original benchmark that contain relevant text elements.

- **Prompt Generation:** For each base image and each target subtask type $s_\Delta$, we generated new prompts focused *exclusively* on performing operations of that type. For example, using an image containing a cat, the prompt for the $\mathcal{D}_{\text{Object Replacement}}$ dataset might be "replace the cat with a dog," while the prompt for the same image in the $\mathcal{D}_{\text{Object Recoloration}}$ dataset could be "recolor the cat to pink." This ensures that when testing a change related to Object Replacement subroutines, the evaluation focuses solely on the performance of that subtask type.

- **Varying Complexity:** Within each subtask-specific dataset $\mathcal{D}_{s_\Delta}$, the generated tasks feature varying complexity. For instance, in $\mathcal{D}_{\text{Object Removal}}$, some tasks might involve removing only one object, while others might require removing six or seven different objects from the same image. This ensures that subroutines are tested across different levels of difficulty for their specific function.

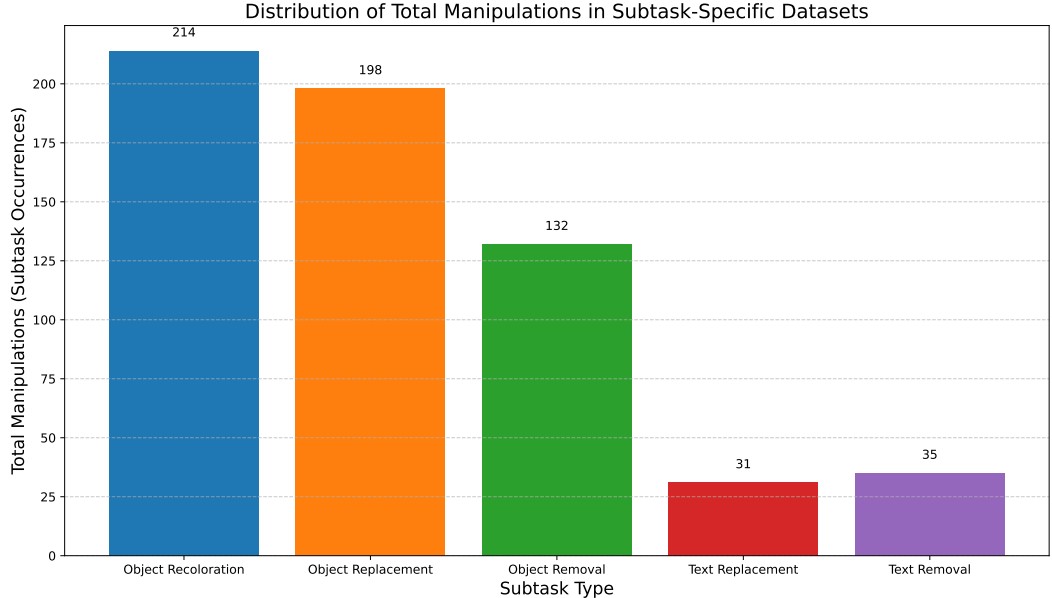

Figure 10: Distribution of total manipulations (subtask occurrences) across the specialized test datasets ($\mathcal{D}_{s_\Delta}$) used for subroutine verification.

- **Dataset Statistics:** Figure 10 shows the distribution of total manipulations (subtask occurrences) for each subtask-specific dataset. While image-based datasets share the same 121 base images, the total number of manipulations can differ based on the number of relevant objects/regions suitable for that subtask type within each image and the varying complexity levels introduced in the prompts.

### L.2 EVALUATION PROTOCOL FOR SUBROUTINE CHANGES

When evaluating a proposed change $\Delta$ for a subroutine related to subtask $s_\Delta$, we follow the procedure outlined in Section G.4:

- **Test Set Sampling ($\mathcal{T}'$, $\mathcal{T}''$):** A random subset of tasks is sampled from the corresponding subtask-specific dataset $\mathcal{D}_{s_\Delta}$. We sample 25 tasks for image-based subtasks and 20 tasks for text-based subtasks for each evaluation task (both baseline and candidate evaluation, including retries).

- **Quality Evaluation ($Q_{base}, Q_{new}$):** The quality score used for calculating the Net Benefit $B(\Delta)$ relies on automated VLM checks. For each task in the sampled test set $\mathcal{T}'$ (or $\mathcal{T}''$), we execute the respective system (baseline CoSTA*/FaSTA* or FaSTA* with the candidate rule change $\mathcal{R}'$). During execution, the VLM quality check is applied after relevant tool steps, using the same methodology as used in CoSTA*. The quality score for a single task is computed as the average of the VLM quality scores obtained for all its constituent subtasks (which are all of type $s_\Delta$ in these specialized datasets). The final quality metric ($Q_{base}$ or $Q_{new}$) used in the Net Benefit calculation is the average of these task-level quality scores across all tasks sampled in $\mathcal{T}'$ (or $\mathcal{T}''$).

- **Cost Evaluation ($C_{base}, C_{new}$):** The cost metric is the average total execution time (in seconds) across all tasks sampled in the test set $\mathcal{T}'$ (or $\mathcal{T}''$).

This detailed dataset construction and evaluation protocol allows for a focused and rigorous assessment of the impact of proposed subroutine changes on performance for the specific subtask type they target.

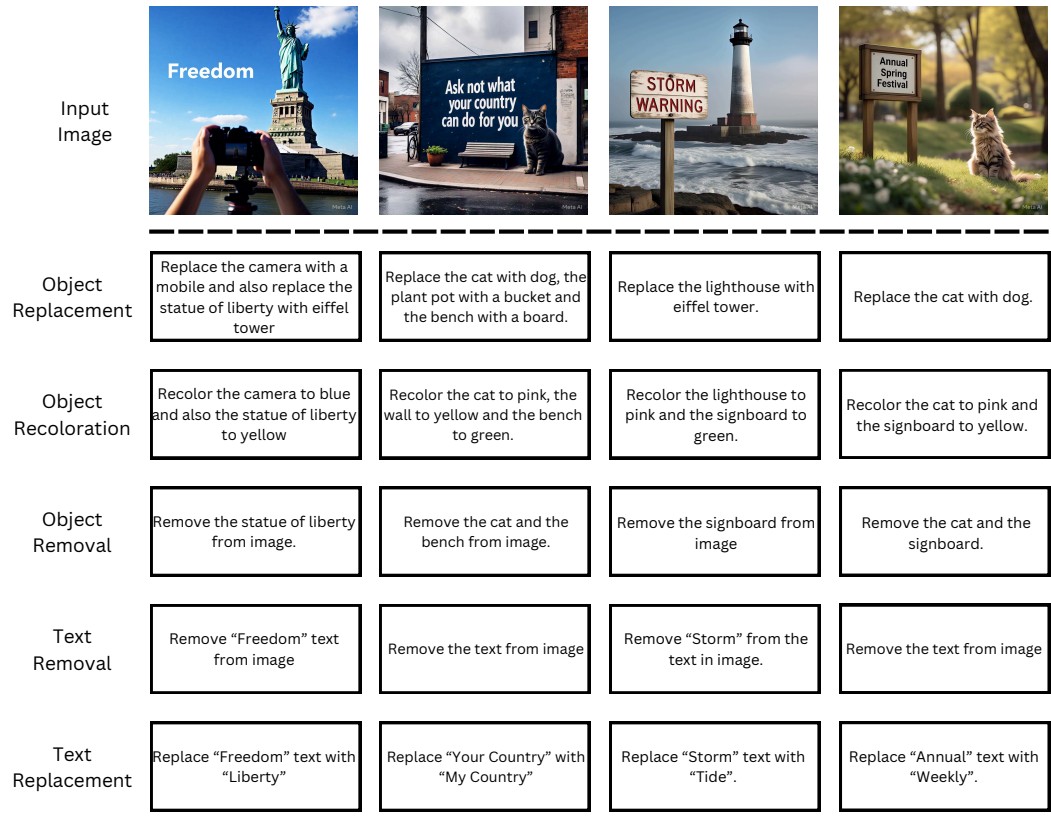

Figure 11: Illustration of subtask-specific dataset generation. A single base image from the CoSTA* benchmark is used with different prompts, each targeting a distinct subtask type, to create evaluation instances for different datasets (e.g., $\mathcal{D}_{\text{Object Replacement}}$, $\mathcal{D}_{\text{Object Recoloration}}$).

## M    TRACE DATA FOR INDUCTIVE REASONING

The online subroutine induction and refinement process (Section 4.2) relies on analyzing execution traces to identify patterns and propose subroutine rules. This appendix details the composition of these traces and provides an example.

### M.1    TRACE COMPOSITION AND DATA GATHERING

For each task processed by FaSTA*, a detailed trace $\tau$ is logged. This trace is crucial for the LLM to perform inductive reasoning during the periodic refinement phase. The key information captured in a trace for each subtask within a completed task includes:

- **Subtask** ($s_j$)**:** The specific subtask being performed (e.g., Object Recoloration, Text Removal).
- **Chosen Tool Path** ($\mathcal{P}_j$)**:** The sequence of tools that was actually executed to complete the subtask (this could be a selected subroutine or a path found via low-level A search).
- **Context Features:** A set of relevant features extracted from the image context and the state of the objects being manipulated. This information can be gathered upfront for the relevant objects in the image or also during the execution of different tools along the path depending on the specific detail.
    - **Object-Specific Features (List not exhaustive):**
        ○ object_size: Derived from bounding boxes provided by object detection tools like YOLO (Wang et al., 2022) (or Grounding DINO (Liu et al., 2024) if YOLO (Wang et al., 2022)class is not supported for a primary object).

- ○ `mask_properties`: Such as mask area or mask ratio, obtained from segmentation tools like SAM (Kirillov et al., 2023).
      - ○ `color_details`: The color of the original object, also derived from the mask output by SAM.
   – **Text-Specific Features:**
      - ○ `text_box_size`: Obtained from text detection tools like CRAFT (Baek et al., 2019).
   – **Relational/Global Features (Queried from LLM):** For each primary object involved in the task, we also query an LLM at the beginning of the task to infer higher-level contextual attributes based on the image and prompt. This is done once for all objects. Examples include:
      - ○ `background_content_type`: Describes the area surrounding or behind an object (e.g., "Simple_Texture", "Homogenous_Area", "Complex_Scene").
      - ○ `overlapping_critical_elements`: A boolean indicating if an object overlaps with other elements (text, other objects) that are targets in subsequent subtasks.
      - ○ `object_clarity`: (e.g., "High", "Medium", "Low" - indicating how clearly visible and unambiguous the object is).

- **Path Cost** ($C_{path}$): The aggregated execution cost (e.g., time) for the chosen tool path $\mathcal{P}_j$.
- **Path Quality** ($Q_{path}$): The aggregated quality score for $\mathcal{P}_j$, an average of VLM scores for the tools in the path.
- **Failures:** Information about any tool failures or VLM quality check failures encountered during the execution of $\mathcal{P}_j$, along with the specific context features active at the time of failure.

This comprehensive trace data, particularly the context features gathered from tools like YOLO, SAM, CRAFT (Baek et al., 2019), and initial LLM queries, allows the inductive reasoning LLM (Section 4.2, Step 3) to correlate observed conditions with the success or failure of different tool paths, thereby proposing or refining subroutines and their activation rules.

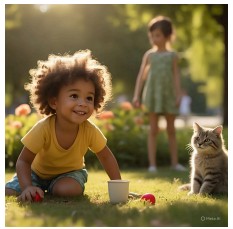

| | |
|---|---|
| **Path**: Grounding DINO → SAM → SD Inpaint | **Path**: SD Search&Recolor |
| **Subtask**: Object Recoloration | **Subtask**: Object Recoloration |
| **Object**: Cup | **Object**: Cup |
| **Object Size**: x1 (From YOLO) | **Object Size**: x1 (From YOLO) |
| **Mask Size**: x2 (From SAM) | **Mask Size**: x2 (From SAM) |
| **Original Color**: <RGB Values> (From SAM) | **Original Color**: <RGB Values> (From SAM) |
| **Target Color**: Blue (From Prompt) | **Target Color**: Blue (From Prompt) |
| **Overlapping Critical Elements**: None (From LLM) | **Overlapping Critical Elements**: None (From LLM) |
| **YOLO Class Support**: Yes | **YOLO Class Support**: Yes |
| **Path Status**: Fail | **Path Status**: Success |

Input Image      Initial Failed Toolpath      Final Successful Toolpath

Figure 12: Visual example for the object recoloration trace detailed in Appendix M.2. Left: Input image. Right: Conceptual representation of the initial failed toolpath trace and the subsequent successful toolpath trace with key context features noted.

## M.2 EXAMPLE TRACE FOR AN OBJECT RECOLORATION SUBTASK

Consider an input image and prompt as shown in Figure 12. The task is "Recolor the cup to blue." The subtask chain might simply be 'Object Recoloration (ball -> blue ball)' with an initial path (Grounding DINO (Liu et al., 2024) -> SAM (Kirillov et al., 2023)-> SD Inpaint (Rombach et al., 2022)) where 'SD Inpaint' failed quality check so path was retraced and a final path (SD Search&Recolor (Rombach et al., 2022)) which passed all quality checks.

The logged traces for this subtask are shown on right side in Figure 12. All details extracted from various tools used along the path, such as YOLO (Wang et al., 2022)and SAM, are included along with the status of the path. Some extra details which are not possible to be extracted from these tools are extracted with the help of an LLM which is called at the start of the task for all related objects (in this case only the ball). In case of the paths where tools like the YOLO (Wang et al., 2022)or SAM (Kirillov et al., 2023)are not included like in case of 'SD Search&Recolor', we use the LLM to extract an approximation of size, color, etc. information as well along with the other extra details for which this LLM was already needed.

This structured trace provides rich data for the LLM to analyze during the subroutine induction and refinement phase.

## N    SUBROUTINE REUSE RATE

Figure 13 illustrates the reuse rate for each of the learned subroutines. The reuse rate is a critical metric that quantifies the utility and applicability of each mined subroutine. Specifically, for each subroutine, this rate is calculated as the percentage of applicable subtasks where that particular subroutine was selected for execution and was also executed successfully. This means the calculation considers only those instances where a subtask was of a type that the subroutine could address. For example, if a subroutine is designed for "Object Recoloration", its reuse rate is determined by how often it was chosen when an "Object Recoloration" subtask was encountered, irrespective of the total number of other subtask types (e.g., "Object Removal", "Text Replacement") processed. A higher reuse rate indicates that a subroutine is frequently chosen when it is relevant, underscoring its effectiveness and the successful learning of its activation rules. The figure displays these rates for all sixteen subroutines (SR1 through SR16), providing insight into their individual contributions to the agent's efficiency.

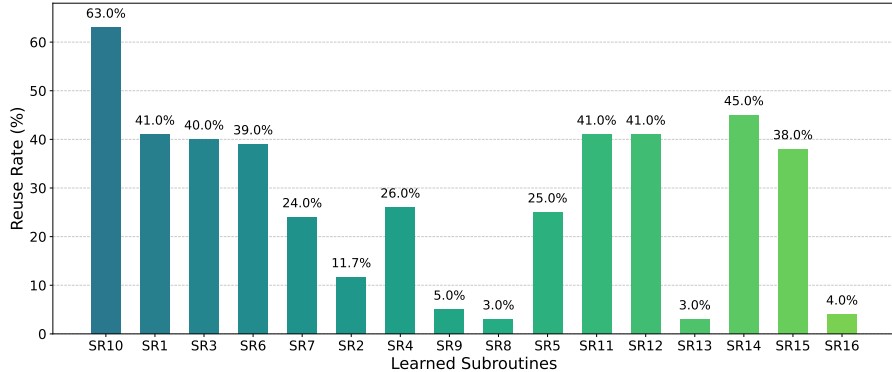

Figure 13: Reuse rate (%) for all learned subroutines. The rate for each subroutine is calculated based on the percentage of applicable subtasks where it was selected for execution.

## O USE OF LARGE LANGUAGE MODELS (LLMS)

It must be noted that LLMs were not used for writing this paper from scratch. However, we use LLMs to polish the content. Also, they do make up fundamental components of the FaSTA* architecture. Their roles are integral to the methodology as has been explained below:

1. **Subtask Chain Generation:** The initial stage of our fast-planning pipeline relies on an LLM to interpret the user's multimodal input (an image and a natural language prompt). We use GPT-4o for this purpose. The LLM decomposes the user's complex, high-level instruction into a logical sequence of discrete editing operations, which we call a *subtask chain*. This process transforms a high-level user request into a structured, executable plan. The specific prompt used to guide the LLM in this task decomposition stage is provided in Appendix Q.

2. **Subroutine Selection:** The second stage of the "fast planning" mechanism uses GPT-o1. For each step in the subtask chain, the LLM refers to the Subroutine Rule Table, which is dynamically updated and the current image context to select the most appropriate, cost-effective subroutine. If no suitable subroutine is found, the system falls back to the "slow planning" A* search. The prompt used for this process is provided in Appendix R.

3. **Online Subroutine Induction:** One of the core contributions of FaSTA* is its ability to learn from past experiences. This learning is driven by an LLM that performs inductive reasoning on execution traces from previously completed tasks. The LLM analyzes logs of successful and failed tool paths, along with their associated contextual features, to identify recurring, efficient patterns. From these patterns, it synthesizes compact, symbolic rules that define reusable subroutines and their activation conditions. This process is detailed in Section 4.2, and the prompt for this symbolic reasoning task can be found in Appendix S.

4. **Quality Verification:** The quality score used to assess the success or failure status of an execution is calculated using VLMs. The VLM check is applied after each tool step, where the VLM is asked to score the editing operation for a particular tool. If this quality check fails, FaSTA* falls back to the slow-planning A* search.

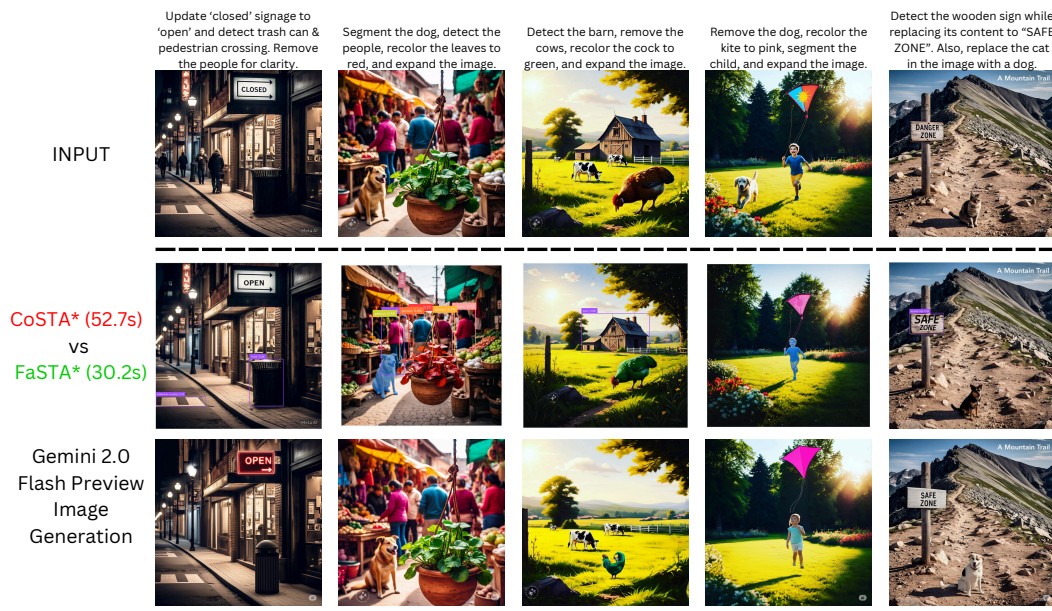

Figure 14: Qualitative comparison of FaSTA* against CoSTA* (Gupta et al., 2025) and Gemini 2.0 Flash Preview for complex multi-turn editing tasks (inputs on top). FaSTA* produces high-quality outputs visually identical to CoSTA* and superior to Gemini. Notably, FaSTA* achieves this CoSTA*-level quality at a significantly reduced execution cost, demonstrating its enhanced efficiency.

# P  ALGORITHMS

## P.1  FASTA* ONLINE SUBROUTINE INDUCTION

---

**Algorithm 1:** High-Level Overview of Online Subroutine Learning.

---

1  Log execution traces (data, paths, outcomes, context) continuously
2  After $K = 20$ task executions, using recent traces:
3      LLM analyzes traces to propose/refine subroutines & activation rules
4      Validate proposals on test data for net cost-quality benefit
5      **if** *validation fails* **then** LLM refines proposal with feedback (max $N_{retries} = 2$ attempts)
6      **else if** *validation succeeds* **then** Update Subroutine Rule Table $\mathcal{R}$ with beneficial change
7  Repeat indefinitely to adapt and improve

---

**Algorithm 2:** FaSTA* Online Subroutine Induction (Modified ICRL). Teal steps are specific additions/modifications for FaSTA*.

---

**Input:** LLM $\pi_{LLM}$, Planner FaSTA*, Initial Rule Set $\mathcal{R}_0$, Trace Buffer $\mathcal{B}$, Update Freq. $K$, Max Retries $N_{retries}$, Subtask Test Datasets $\{\mathcal{D}_s\}_{s \in S}$
**Output:** Continuously Updated Rule Set $\mathcal{R}$

1  $\mathcal{R} \leftarrow \mathcal{R}_0$;
2  $\mathcal{B} \leftarrow \emptyset$;
3  $t \leftarrow 0$;
4  **while** *True* **do**
5      Receive new task input $(x_t, u_t)$;
6      Execute FaSTA* with current rules $\mathcal{R}$ to get trace
        $\tau_t = (\text{subtask}, \mathcal{P}_{\text{final}}, \text{ContextFeatures}, C_{\text{path}}, Q_{\text{path}}, \text{Failures})$;
7      Add $\tau_t$ to Buffer $\mathcal{B} \leftarrow \mathcal{B} \cup \{\tau_t\}$;
8      $t \leftarrow t + 1$;
9      **if** $t > 0$ **and** $t \pmod{K} == 0$ **then**
10          $\mathcal{T}_{recent} \leftarrow$ Get last K traces from $\mathcal{B}$;
11          $\Delta_{proposals} \leftarrow \pi_{LLM}(\text{"Propose changes to rules } \mathcal{R} \text{ based on recent traces } \mathcal{T}_{recent}\text{"})$;
12          **foreach** *proposed_change* $\Delta$ *in* $\Delta_{proposals}$ **do**
13              $s_\Delta \leftarrow$ GetSubtaskType($\Delta$);
14              accepted $\leftarrow$ False;
15              current_delta $\leftarrow \Delta$;
16              **for** $retry \leftarrow 0$ **to** $N_{retries}$ **do**
17                  $\mathcal{T}_{test} \leftarrow$ SampleRandomSubset($\mathcal{D}_{s_\Delta}$);
18                  **if** $t == K$ **then**
19                      $(C_{base}, Q_{base}) \leftarrow$ Evaluate(CoSTA*, $\mathcal{T}_{test}$);
20                  **else**
21                      $(C_{base}, Q_{base}) \leftarrow$ Evaluate(FaSTA* ($\mathcal{R}$), $\mathcal{T}_{test}$);
22                  $\mathcal{R}_{candidate} \leftarrow$ ApplyChange($\mathcal{R}$, current_delta);
23                  $(C_{new}, Q_{new}) \leftarrow$ Evaluate(FaSTA* ($\mathcal{R}_{candidate}$), $\mathcal{T}_{test}$);
24                  $\Delta C\% \leftarrow (C_{new} - C_{base})/C_{base} \times 100$;
25                  $\Delta Q\% \leftarrow (Q_{new} - Q_{base})/Q_{base} \times 100$;
26                  $B(\Delta_{eval}) \leftarrow \Delta C\% - \Delta Q\%$;
27                  **if** $B(\Delta_{eval}) < 0$ **then**
28                      $\mathcal{R} \leftarrow \mathcal{R}_{candidate}$;
29                      accepted $\leftarrow$ True;
30                      Break;
31                  **else if** $retry < N_{retries}$ **then**
32                      feedback $\leftarrow$ AnalyzeFailure(current_delta, $C_{new}, Q_{new}, C_{base}, Q_{base}$, $\mathcal{T}_{test}$);
33                      current_delta $\leftarrow \pi_{LLM}(\text{"Refine change } \Delta \text{ based on feedback: feedback"})$;
34          // Standard ICRL might sample past episodes here to build context for next action (e.g., Algo 1/2 in (Monea et al., 2025)). FaSTA* replaces this with explicit rule update.

## P.2 FASTA* ADAPTIVE FAST-SLOW EXECUTION

---

**Algorithm 3:** High-Level Overview of Adaptive Fast-Slow Planning. Detailed Algorithm can be found in Appendix P.2.

---

```
// Input:   Subtask chain, Subroutine Rule Table R
```
1 LLM generates a "Fast Plan" (sequence of subroutines or 'None') for the subtask chain
2 **foreach** *step in the Fast Plan* **do**
3      **if** *Fast Plan step is a valid subroutine* **then**
4          Attempt to execute the subroutine
5          **if** *subroutine fails VLM quality check* **then** trigger Slow Path for this subtask
6      **else**
7          Trigger Slow Path for this subtask
8      **if** *Slow Path was triggered* **then**
9          Perform localized A* search on the subtask's low-level subgraph
10      Proceed to next step upon successful completion of current subtask

---

**Algorithm 4:** FaSTA* Adaptive Fast-Slow Execution

---

**Input:** Input image $x_0$, Prompt $u$, Subtask Chain $G_{sc}$, Subroutine Rule Table $\mathcal{R}$, LLM $\pi_{LLM}$, VLM, Quality Threshold $Q_{thresh}$, Cost params $\alpha$, BT, MDT, TDG

**Output:** Final Edited Image $x_{final}$

1 Generate Fast Plan $\mathcal{M}_{subseq} \leftarrow \pi_{LLM}(x_0, u, G_{sc}, \mathcal{R})$;
2 Current Image $x_{curr} \leftarrow x_0$;
3 Path Trace $\tau_{path} \leftarrow []$;
4 **foreach** *subtask $s_i$ in sequence from $G_{sc}$* **do**
5      plan_step $\leftarrow \mathcal{M}_{subseq}(s_i)$;
6      subtask_success $\leftarrow$ False;
7      **if** *plan_step is a Subroutine $\mathcal{P}^*_{s_i}$* **then**
         /* Attempt Fast Plan                                 */
8          temp_image $\leftarrow x_{curr}$; fast_path_ok $\leftarrow$ True; subroutine_trace $\leftarrow []$;
9          **foreach** *tool $t_k$ in $\mathcal{P}^*_{s_i}$* **do**
10              Execute $t_k$ on temp_image to get $x_{int}$, cost $c_k$;
11              Append $(t_k, c_k)$ to subroutine_trace;
12              $q_k \leftarrow$ VLMQualityCheck$(x_{int}, s_i, t_k)$;        // Per (Gupta et al., 2025)
13              **if** *$q_k < Q_{thresh}$* **then**
14                  fast_path_ok $\leftarrow$ False; Break;
15              temp_image $\leftarrow x_{int}$;
16          **if** *fast_path_ok* **then**
17              $x_{curr} \leftarrow$ temp_image;
18              Append subroutine_trace to $\tau_{path}$;
19              subtask_success $\leftarrow$ True;
20      **else**
         /* plan_step is 'None'                                   */
21      **if** *not subtask_success* **then**
         /* Execute Slow Path (Local A*)                     */
22          $G_{low}(s_i) \leftarrow$ ConstructSubgraph$(s_i, $MDT, TDG$)$;      // Per (Gupta et al., 2025)
23          slow_trace $\leftarrow$ LocalAStarSearch$(G_{low}(s_i), x_{curr}, s_i, \alpha, $BT, VLM, $Q_{thresh})$;
24          $x_{curr} \leftarrow x_{out}$;
25          Append slow_trace to $\tau_{path}$;
26          subtask_success $\leftarrow$ True;
27 Return $x_{curr}$, $\tau_{path}$;

---

# Q LLM PROMPT FOR GENERATING SUBTASK CHAIN

**You are an advanced reasoning model responsible for decomposing a given image editing task into a structured subtask chain. Your task is to generate a well-formed subtask chain that logically organizes all necessary steps to fulfill the given user prompt. Below are key guidelines and expectations:**

## Q.1 UNDERSTANDING THE SUBTASK CHAIN

A subtask chain is a structured representation of how the given image editing task should be broken down into smaller, logically ordered subtasks. Each node in the chain represents a subtask which is involved in the prompt, and edges represent the ordering like which subtask needs to be completed before or after which.
Each node of the chain represents the subtasks required to complete the task. The chain ensures that all necessary operations are logically ordered, meaning a subtask that depends on another must appear after its dependency.

## Q.2 STEPS TO GENERATE THE SUBTASK CHAIN

- **Step 1:** Identify all relevant subtasks needed to fulfill the given prompt.
- **Step 2:** Ensure that each subtask is logically ordered, meaning operations dependent on another should be placed later in the path.
- **Step 3:** Each subtask should be uniquely labeled based on the object it applies to and of the format (Obj1 → Obj2) where obj1 is to be replaced with obj2 and in case of recoloring (obj → new color) while with removal just include (obj) which is to be removed. Example: If two objects require replacement, the subtasks should be labeled distinctly, such as `Object Replacement (Obj1 -> Obj2)`.
- **Step 4:** There also might be multiple possible subtasks for a particular requirement like if a part of task is to replace the cat with a pink dog then the two possible ways are `Object Replacement (cat-> pink dog)` and another is `Object Replacement (cat->dog) -> Object Recoloration (dog->pink)`

## Q.3 LOGICAL CONSTRAINTS & DEPENDENCIES

When constructing the chain, keep in mind that you take care of the order as well like if a task involves replacing an object with something and then doing some operation on the new object then this operation should always be after the object replacement for this object since we cannot do the operation on the new object till it is actually created and in the image.

## Q.4 INPUT FORMAT

The LLM will receive:

1. An image.
2. A text prompt describing the editing task.
3. A predefined list of subtasks the model supports (provided below).

## Q.5 SUPPORTED SUBTASKS

Here is the complete list of subtasks available for constructing the subtask chain: Object Detection, Object Segmentation, Object Addition, Object Removal, Background Removal, Landmark Detection, Object Replacement, Image Upscaling, Image Captioning, Changing Scenery, Object Recoloration, Outpainting, Depth Estimation, Image Deblurring, Text Extraction, Text Replacement, Text Removal, Text Addition, Text Redaction, Question Answering based on text, Keyword Highlighting, Sentiment Analysis, Caption Consistency Check, Text Detection
**You must strictly use only these subtasks when constructing the chain.**

## Q.6 EXPECTED OUTPUT FORMAT

The model should output the subtask chain in structured JSON format, with each node having:

- **Subtask Name** (with object label if applicable)
- **Parent Node** (Parent node of that subtask)
- **Execution Order** (logical flow of tasks)

## Q.7 EXAMPLE INPUTS & EXPECTED OUTPUTS

### Q.7.1 EXAMPLE 1

**Input Prompt:** *"Detect the pedestrians, remove the car and replacement the cat with rabbit and recolor the dog to pink."*

**Expected Subtask chain:**

```
{
    "task": "Detect the pedestrians, remove the car and

replacement the cat with rabbit and recolor the dog to pink",

    "subtask_chain": [
        {
            "subtask": "Object Detection (Pedestrian)(1)",
            "parent": []
        },
        {
            "subtask": "Object Removal (Car)(2)",
            "parent": ["Object Detection (Pedestrian)(1)"]
        },
        {
            "subtask": "Object Replacement (Cat -> Rabbit)(3)",
            "parent": ["Object Removal (Car)(2)"]
        },
    {
        "subtask": "Object Recoloration (Dog -> Pink Dog)(4)",
        "parent": ["Object Replacement (Cat -> Rabbit)(3)"]
    }
    ]
}
```

### Q.7.2 EXAMPLE 2

**Input Prompt:** *"Detect the text in the image. Update the closed signage to open while detecting the trash can and pedestrian crossing for better scene understanding. Also, remove the people for clarity. "*

**Expected Subtask chain:**

```
{
    "task": "Detect the text in the image. Update the closed

     signage to open while detecting the trash can and

     pedestrian crossing for better scene understanding. Also,

     remove the people for clarity.",

    "subtask_chain": [
        {
            "subtask": "Text Replacement (CLOSED -> OPEN)(1)",
            "parent": []
        },
        {
        "subtask": "Object Detection (Pedestrian Crossing)(2)",
        "parent": ["Text Replacement (CLOSED -> OPEN)(1)"]
        },
```

```
        {
        "subtask": "Object Detection (Trash Can)(3)",
        "parent": ["Object Detection (Pedestrian Crossing)(2)"]
        },
        {
            "subtask": "Object Removal (People)(4)",
            "parent": ["Object Detection (Trash Can)(3)"]
        },
        {
            "subtask": "Text Detection ()(5)",
            "parent": ["Object Removal (People)(4)"]
        }
    ]
}
```

*You can observe in the second example since there was a subtask related to text replacement, it made sense to detec the text at last after all changes to text had been made. You should always be mindful that ordering is logical and if there is a subtask whose output or input might change based on some other subtask's operation then it is always after this subtask on whose operation it depends. eg- "recolor the car to pink and replace the truck with car" so in this one the recoloration of car always depends on the replecement of truck with car so the recoloration should always be done after replacement so you should think logically and it is not necessary that the sequence of subtasks in subtask chain is same as they are mentioned in the input prompt as was in this case the recoloration was mentioned before replacement in input prompt but logically replacement will come first.*

### Q.8  YOUR TASK

**Now, using the given input image and prompt, generate a well-structured subtask chain that adheres to the principles outlined above.**

- Ensure logical ordering and clear dependencies.
- Label subtasks by object name where needed.
- Structure the output as a JSON-formatted subtask chain.

**Input Details**

- Image: `input image`
- Prompt: ["prompt"]
- Supported Subtasks: (See the list above)

**Now, generate the correct subtask chain.** *Before you generate the chain you need to make sure that for every path possible in the subtask chain all the subtasks in that chain are covered and none are skipped. Also if a prompt involves something related to object replacement then just have that you dont need to think about its prerequisites like detecting it or anything bcz it already covers it.*

## R    LLM PROMPT FOR SUBROUTINE SELECTION

So we have this image: `<image>` and also have the following input prompt:
`<prompt>`
So we got the following subtask chain:
`<subtask chain>`
Note that in the subtask chain within a particular node there is a bracket which tell us about the object from and target and if there is only from then target is not mentioned like in removal only from object is needed no target is required while for replacement/recoloration target is also required.
Now we have the following subroutines list for each subtask and each of the subroutines have some observations related to them which specify under which conditions they are to be used or not. So you need to read those subroutines and their observations then check the corresponding object for that subtask within the image like if its `Object Removal (Cat)` then check the cat in image and then from the subroutines list check that if for that particular subtask there is any subroutine in which the observation conditions are satisfied and if so give the list of those subroutines for that subtask and you need to do this for all subtasks in the subtask chain.
Subroutine list and the details:
`{Subroutine Rule Table}`

EXAMPLE:

Suppose we have an image which has lots of objects along with a very large car which has a background with lots of objects and also a brown wooden board with some text written on it. Now we have a prompt that remove the car and recolor the wooden board to pink and detect the text and get the following subtask chain:
`Object Removal (Car) -> Object Recoloration (Wooden Board -> Pink Wooden Board) -> Text Detection ()`
Now we see the subroutine list and find that for removal since the object is too big sub7 and sub8 are not possible. Now in sub9 and sub10 we see that the 'car' class is supported by yolo so eventually we choose sub10 for this subtask. For recoloration we see that it has object (text) which is imp and is involved in subsequent subtask so sub1 and sub3 aren't possible and we see that the color of board is light brown so light brown and pink dont have too much difference so we choose sub2. For text detection there is not subroutine available so we leave it like that.
So output will be:

```
Object Removal (Car) : [sub10]
Object Recoloration (Wooden Board -> Pink Wooden Board) : [sub2]
Text Detection () : [None]
```

Now lets say the wooden board was black in color and had to be recolored to white. In this case the sub1 and sub3 are not possible because of the text as before but now sub2 is also not possible because the color difference is too much. So we do not choose any subroutine for this subtask and output is as follows:

```
Object Removal (Car) : [sub10]
Object Recoloration (Wooden Board -> Pink Wooden Board) : [None]
Text Detection () : [None]
```

Now lets change the details further. Lets say that the wooden board does not have any text written on it and has to be recolored from pink to yellow and the text detection subtask wasn't present so in this case for recoloration all subroutines are possible except sub3 bcz wooden board isnt a class supported by yolo.
New output:

```
Object Removal (Car) : [sub10]
Object Recoloration (Wooden Board -> Pink Wooden Board) : [sub1,
sub2]
```

Now lets change it a bit assume that all conditions are as original but the car is small and behind the car only some walls, grass, etc are present some basic stuff and not a lot of objects like occluded people, cats, etc so in this case we will choose sub8 and sub10 for it as it is not too plain that sub10 cannot be used and it is not way too complex that sub8 cannot be used. New output:

```
Object Removal (Car) : [sub8, sub10]
Object Recoloration (Wooden Board -> Pink Wooden Board) : [sub2]
Text Detection () : [None]
```

Now you need to do the same things for the current case where the input prompt is : `<prompt>` Subtask chain: `<subtask chain>`

Also multiple options are possible if they satisfy all the conditions it is not necessary that only one is chosen and it is also possible that no subroutine fulfills all conditions so in that case choose None so that we can do A$^*$ search and find the correct output. Also keep in mind that only look at the details relevant say you need to check subroutine for some object which is to be removed and for some activation condition you need to see if the background is busy or plain, etc so you only see the background relevant like near that object and not for the entire image.

So you need to extract all relevant details related to all relevant objects from the image given to you then check the subroutine list if anyone matches and give the output.

## S    LLM PROMPT FOR INDUCTIVE REASONING ON SUBROUTINES

**Goal:** Analyze the provided experimental run data for a specific task (e.g., Object Recoloration) to infer initial, potentially qualitative, activation rules (preference conditions) for each distinct execution path employed.

`<All Logged Data (The Traces)>`

So we run different models and tools for different or same image editing tasks and store the observations including what path was finally used and what were the conditions of objects, etc. and this data is provided to you. Now we wish to infer some subroutines or commonly used paths and their activation rules under which they are commonly activated. Can you find some commonly used subroutines or paths and infer some rules for these paths using the status of these cases and other factors and give the rules for both paths and they need not be too specific but a bit vague is fine like if you observe that some particular path always fails in case object size is less then you can give the rule that this path should be used when object is not too small and not give any specific values so activation rule will include like `object_size = not too small`, etc like this based on all factors like object size, color transitions, etc and also it is possible that for some path it failed bcz of some specific condition like its not necessary all conditions led to failure so you need to check which is the condition which always leads to failures or which always leads to success and that will constitute a rule if some condition leads to both failures and success with same value then it means that this is not the contributing factorand there's something else that's causing the failure or success and keep in mind that output rules should be of activation format like in what cases this should be used and not negative ones so if there is some path which always fails when object size is big then your activation rule will have `object_size = small` and not some deactivation rules which has `object_size = big`. You should also include some explanatory examples in the rule which can help some person or LLM understand them better when referring to these rules. eg. if there is a rule where you want to say that this path will only succeed when the difference between size of objects is not too big then you can have a rule like : "`size_difference(original, target objects) = Not too big (eg.  hen to car, etc)`" where you include some example. You should focus on activation rules which are like in what case this particular path will always succeed and some activation rules should also include a kind of deactivation rule with a not like in case you observe that some path always fails when there is some condition x where x can be like object is too small or color difference is huge then you should infer an activation rule that is negate of this like the rule can be object is "not" too small or color difference is "not" huge so that these activation rules can act as a kind of deactivation rules as well and prevent the path from getting activated in cases where we know for sure it'll fail.

AN EXAMPLE:

**Experimental Data:**

```
Subtask: s1, Object Size: 0.7px, Original Object Color: Yellow,
 Target Object Color: Black
Path used: P1
Status: Fail

Subtask: s1, Object Size: 0.2px, Original Object Color: Yellow,
 Target Object Color: Green
Path used: P1
Status: Success

Subtask: s1, Object Size: 5px, Original Object Color: White,
Target Object Color: Black
Path used: P1
Status: Fail
```

```
Subtask: s1, Object Size: 5px, Original Object Color: White,
Target Object Color: Yellow
Path used: P1
Status: Success

Subtask: s1, Object Size: 0.7px, Original Object Color: Black,
Target Object Color: White
Path used: P1
Status: Fail

Subtask: s1, Object Size: 3px, Original Object Color: Black,
Target Object Color: Yellow
Path used: P2
Status: Success

Subtask: s1, Object Size: 0.9px, Original Object Color: Black,
Target Object Color: Blue
Path used: P2
Status: Fail

Subtask: s1, Object Size: 0.2px, Original Object Color: White,
Target Object Color: Yellow
Path used: P2
Status: Fail

Subtask: s1, Object Size: 0.6px, Original Object Color: White,
Target Object Color: Blue
Path used: P3
Status: Success
```

So we see that paths P1 and P2 are very commonly used so these will be our subroutines or commonly used paths and now our goal is to infer some rules under which these subroutines or paths are commonly activated. So by observing the data you see that in P2 it always fails when object size is small while the color transitions doesn't matter so for P2 you can infer an activation rule which is "`object_size:  not too small`" and while for P1 you observe that the object size doesn't really matter bcz it is able to succeed in both small and big object sizes and also fail in both cases but you observe that when the color transition is huge like white to black or black to white it always fails while when color transition is not extreme like white to yellow or yellow to green it is able to succeed even under same size conditions so you can infer a rule that it depends on color transition and give a rule with example for better understanding like: "`color_transition:  not too extreme (eg.  not white <-> black, etc.)`"

The real experimental data will include much more info and it is your job to infer what data is useful and find patterns in it and give corresponding rules. Also you should not mix the observations from different paths or subtasks and treat all paths and subtasks independently so while infering rules for some path P1 for some subtask s1 then only look at the experimental data of that path P1 and subtask s1 and infer rules and patterns from that bcz observations of P2 doesn't affect P1 and neither do observations for P1 but related to s2 affect P1 for s1. So you should know that same path can be used for different subtasks and can have different activation rules for different subtasks so while inferring these rules you should see that you compare the object conditions for the same subtask and same path and then reach a final conclusion like example you have some path p1 which is used in subtasks s1 and s2

and based on observations there are multiple failure cases for p1 where object size is small and subtask is s1 while there are some success cases for same p1 where object size is big and subtask is s2 so if you combine them you won't be infer any rule bcz nothing's epcific but you need to treat both the p1's independently one is p1 with s1 and another is p1 with s2 and so for p1 with s1 you can infer a rule that it oonly works when object size is not small.

**The output format for each path for which you can infer some rule/s will be following:**

```
Path: {path1}
Subtask: {subtask}
Activation Rules:
* Rule a1 with some explanatory example if needed
* Rule a2 with some explanatory example if needed
.
* Rule aN with some explanatory example if needed

Path: {path2}
Subtask: {subtask}
Activation Rules:
* Rule b1 with some explanatory example if needed
* Rule b2 with some explanatory example if needed
.
* Rule bN with some explanatory example if needed
```

