# OpenReview forum: "FaSTA*: Fast-Slow Toolpath Agent with Subroutine Mining for Efficient Multi-turn Image Editing"
_ICLR.cc/2026/Conference — ICLR 2026 Poster_

### Official Review · Reviewer_39Yj · 2025-10-20

**Soundness:** 2
**Presentation:** 3
**Contribution:** 2
**Rating:** 4
**Confidence:** 4

**Summary:**

This paper introduces an AI agent called FaSTA* designed to handle multi-step image editing tasks based on a previous work "CoSTA*". The agent works by combining an LLM, which quickly plans the high-level subtasks, with a more detailed A* search that finds the best sequence of AI tools to accomplish each individual step. The key idea is that the system learns from its successful past actions. It uses an LLM to identify common sequences of tools that work well and saves them as reusable subroutines. This creates a "fast-slow" system where the agent first tries to solve a new task quickly by using these pre-saved subroutines. If these shortcuts don't work for a novel or difficult task, the agent then switches to the slower, more methodical A* search to find a custom solution. The authors show that this approach makes FaSTA* significantly more computationally efficient while maintaining a success rate that is competitive with CoSTA*.

**Strengths:**

1. Despite intuitive, the proposed "fast-slow" strategy is a successful attempt to save inference time.
2. The writing and figures ease the understanding of the motivation and method.
3. The editing results are impressive, and the coverage of editing types is comprehensive.

**Weaknesses:**

1. Limited Technical Contribution. The paper's technical contributions to both the neurosymbolic and agent planning fields appear limited. From the neurosymbolic aspect, the mechanism for learning symbolic "rules" is a straightforward process of summarizing and retrieving successful execution paths. While functional, this approach offers limited insights for rule induction and program synthesis explored in the broader neurosymbolic literature. From the agent planning perspective, the core innovation can be viewed as augmenting the A* search with a retrieval-augmented generation (RAG) step to recall past solutions. While this is a practical engineering choice, it offers limited new insight into the fundamental challenges of agentic planning. I also suggest the authors conduct a more comprehensive review of agents' learning capability and memory, which is very relevant to this work but was unfortunately missed.

2. Potentially Inequitable Experimental Comparison. The experimental comparisons may not provide a fair assessment. The baseline methods were not specifically optimized for the custom multi-turn editing benchmark introduced in CoSTA∗, potentially placing them at a significant disadvantage. A more informative comparison would involve a well-engineered ReAct agent equipped with the exact same toolset and effective prompt engineering. Such a baseline would help isolate whether the observed performance gains stem truly from the proposed fast-slow planning mechanism or from factors common to most modern agentic frameworks. The core logic of CoSTA*, FaSTA*, and a ReAct agent is fundamentally similar.

3. Questionable Problem Framing and Motivation. The paper frames its primary contribution around optimizing for "cost" in multi-turn image editing. However, it is questionable whether inference time is the most critical bottleneck that needs addressing in this domain. Furthermore, as an incremental work building upon the CoSTA* framework, the added contribution, i.e., the learning and reuse of subroutines, feels like a modest step forward rather than a significant leap. The novelty seems insufficient for a standalone publication at ICLR.

4. Minor Point on Formatting. As a minor note, the excessive use of vspace throughout the paper negatively impacts readability. Not sure if this should be desk rejected.

**Questions:**

See weaknesses.

---

> ### Author Response · Authors · 2025-11-23
> **Response to Reviewer 39Yj (1 of 2)**
>
> Thank you for your detailed feedback! We address your comments below.
>
> > Q1: The rule learning is merely path summarization and the core innovation can be viewed as just "A* + RAG." The literature review on agent memory is missing.
>
> FaSTA* is not "A* + RAG": RAG does not cover the **inductive reasoning and online learning of reusable rules**, which is the key to generalization. The importance and difference of this component has been analyzed in **Appendix I.1**, where we compare FaSTA* with a RAG baseline ("Direct Caching") that stores all the explored toolpaths and retrieved the most relevant ones to be included in the agent's input context.
>
> The RAG baseline failed with a **95% fallback rate** because exact matches of traces/toolpaths in pixel/feature space are rare. In contrast, FaSTA*'s inductive rules (e.g., "use Tool A if `object_size` is small") achieved a **9% fallback rate**. This **86% performance gap** demonstrates the advantage of learning a reward verified (Net Benefit in Section 4.2), generalizable **editing policy** vs. recalling past solutions.
>
> We appreciate the suggestion regarding the literature review on agent memory and learning. We will expand our related work section to include a comprehensive review of these areas to better contextualize our neurosymbolic approach.
>
> > **Q2: Baselines lack optimization for the custom benchmark, and a well-engineered ReAct agent with the same toolset is required to isolate the fast-slow mechanism's benefits.**
>
> FaSTA*, CoSTA*, and ReAct have fundamental differences in their methodologies:
> - ReAct agents iterates between reasoning and acting by LLMs.
> - CoSTA* outperforms ReAct agents (like GenArtist and CLOVA) by combining the complementary strengths of LLM reasoning and cost-sensitive A* search, with VLM feedback incorporated.
> - FaSTA* features online learning of reusable subroutines from previous experiences and applies a dynamic library of learned rules to future tasks. So it can achieve progressive improvement shown in Fig. 1. In contrast, ReAct and CoSTA* are pure inference strategies and do not learn general knowledge across tasks/instances.
> - FaSTA* features a novel fast-slow planning agent that calls CoSTA* as a "slow-planning" tool when the fast planning policy (defined by the learned rules) fails.
>
>
> **Fair Comparison with Restricted Toolsets:**
> To strictly address the "inequitable comparison" due to toolset differences, we performed additional ablation studies where **FaSTA\* and CoSTA\* were restricted to exact the same toolsets and subtasks** adopted by the baselines (VisProg, CLOVA, GenArtist).
>
> **Table A: Fair comparison with VisProg and CLOVA using their restricted toolset.** Performance difference (Diff. w/ FaSTA*) is calculated as the average over all baselines vs. FaSTA*.
>
> | Subtasks | CoSTA* | **FaSTA*** | VisProg | CLOVA | Diff. w/ FaSTA* |
> | :--- | :--- | :--- | :--- | :--- | :--- |
> | 1-2 Subtasks | 0.510 | **0.509** | 0.498 | 0.504 | -1.6% |
> | 3-4 Subtasks | 0.551 | **0.549** | 0.489 | 0.496 | -10.3% |
> | 5-6 Subtasks | 0.573 | **0.571** | 0.446 | 0.451 | -21.4% |
> | 7-8 Subtasks | 0.460 | **0.456** | 0.301 | 0.310 | -33.0% |
> | **Overall Accuracy** | **0.525** | **0.521** | **0.436** | **0.446** | **-15.4%** |
>
> **Table B: Fair comparison with GenArtist using GenArtist's restricted toolset.**
> Performance difference (Diff. w/ FaSTA*) is calculated as GenArtist vs. FaSTA*.
>
> | Subtasks | CoSTA* | **FaSTA*** | GenArtist | Diff. w/ FaSTA* |
> | :--- | :--- | :--- | :--- | :--- |
> | 1-2 Subtasks | 0.508 | **0.507** | 0.506 | -0.2% |
> | 3-4 Subtasks | 0.578 | **0.575** | 0.548 | -4.7% |
> | 5-6 Subtasks | 0.606 | **0.602** | 0.503 | -16.4% |
> | 7-8 Subtasks | 0.515 | **0.505** | 0.392 | -22.4% |
> | **Overall Accuracy** | **0.553** | **0.547** | **0.495** | **-9.5%** |
>
> As shown in the above tables, even when handicapped with the restricted toolset, FaSTA* achieves performance comparable to CoSTA* and significantly outperforms the other baselines. This demonstrates that the gains stem from FaSTA*'s' **online learning capabilty and fast-slow planning**, not just a broader toolset.

---

> > ### Author Response · Authors · 2025-11-23
> > **Response to Reviewer 39Yj (2 of 2)**
> >
> > > **Q3:  It is questionable whether inference time is the most critical bottleneck that needs addressing in this domain**
> >
> > Inference latency of multi-turn image editing is a critical bottleneck in both research and industry, particularly for agentic workflows where tool chaining multiplies execution time.
> >
> > Significant efforts of recent research focuses on reducing this cost, such as **"Beta Sampling is All You Need"** (Lee et al., WACV 2025), **"NAMI"** (Ma et al., arXiv 2025), and **"Inverse-and-Edit"** (Beletskii et al., arXiv 2025). This efficiency bottleneck is exacerbated in agentic frameworks because a single task involves a chain of multiple tool calls (e.g., Detection $\to$ Segmentation $\to$ Inpainting). By reducing the total execution cost by ~50% (from ~60s to ~30s), FaSTA* addresses the primary barrier to practical, interactive deployment.
> >
> > >**Q4: The learning and reuse of subroutines, feels like a modest step forward rather than a significant leap. The novelty seems insufficient for a standalone publication at ICLR.**
> >
> > FaSTA* is not an incremental work as it addresses an online learning problem, while CoSTA* focuses on pure inference. FaSTA* calls CoSTA* as an exploration policy and slow-planning tool. More details can be found in our response to Q2.
> >
> > > **Q5: Minor Point on Formatting - excessive use of vspace.**
> >
> > We utilized `vspace` primarily to fit large figures/tables and reduce their blank margins. We will rigorously clean up the formatting and remove any excessive spacing adjustments in the revised version.

---

> > > ### Author Response · Authors · 2025-11-27
> > > **Follow up on the response**
> > >
> > > Dear Reviewer,
> > >
> > >
> > > Thank you again for your time and helpful suggestions. We wanted to follow up to confirm that our responses have fully resolved your concerns. We remain available and are happy to provide any further clarifications should you have additional questions during the remaining discussion period.
> > >
> > >
> > > Sincerely,
> > >
> > > The authors.

---

### Official Review · Reviewer_wGeb · 2025-10-23

**Soundness:** 3
**Presentation:** 2
**Contribution:** 3
**Rating:** 8
**Confidence:** 3

**Summary:**

This paper introduces FaSTA, a neurosymbolic agent designed to significantly improve the computational efficiency of complex, multi-turn image editing. The work addresses the key bottleneck of its predecessors like e.g CoSTA, which rely on computationally expensive A* search for every task without learning from past experiences.

FaSTA*'s core innovation is a novel planning framework that automatically analyzes successful execution traces from previous tasks, using LLMs for inductive reasoning to extract frequently used sequences of tool calls as reusable "subroutines." These subroutines are generalized into symbolic rules with activation conditions (e.g., object size, background complexity).

During execution, FaSTA* first generates a cost-effective "Fast Plan" by having an LLM select appropriate learned subroutines for each subtask. When a subroutine is unavailable or fails a quality check, it triggers a localized A* search for that specific subtask. This adaptive strategy mimics human-like reasoning, prioritizing efficient, learned shortcuts while retaining robust search for novel challenges.

The results demonstrate that FaSTA* achieves a paradigm shift in cost-efficiency. It reduces average execution cost compared to CoSTA* at the price of a slightly reduced output quality. By transitioning from a test-time-only planner to a continuously learning agent, FaSTA* offers a more scalable and practical solution for complex, tool-based image editing, effectively combining the speed of LLM planning with the guaranteed performance of heuristic search.

**Strengths:**

The main strengths of the paper lie in:
-  The proposed approach exhibits a good efficiency at the price of a small reduction of the quality of the output.

- The "fast-slow" planning paradigm is intuitive and powerful, effectively mimicking human problem-solving.

-  Moves beyond one-off planning to a continuously learning system, turning past costly searches into future time savings.

- Comprehensive experiments and ablations validation of the approach, clearly demonstrating the contribution of each component.

- Provides a clear path towards more affordable and scalable AI tool-use agents for complex tasks.

**Weaknesses:**

The main weaknesses of this work are the following:
- The performances are suboptimal initially, requiring a "warm-up" period of task execution before the subroutine library becomes effective.

- The entire framework hinges on capable (and often expensive) LLMs for planning, subroutine selection, and induction, creating a dependency and potential cost center.

- Builds upon the already complex CoSTA* framework, making the overall system sophisticated and potentially challenging to reproduce, although the code is avaiable at least for reviewing purposes.

- The success of the "Fast Plan" relies on the generality of the learned symbolic rules, which may not capture all edge cases or nuanced visual contexts, and if the learning is bad, the results are properly of bad quality.

- The paper is too short and not self contained. Indeed, without the additional material it is very hard to understand and assess the details. From reading the paper it is also difficult to reproduce the results. For reviewing purposes the code is available, and we recommend to make t availabe if the paper will be accepted.

**Questions:**

1. The paper states performance improves with more tasks, but could you provide a concrete curve or table showing the relationship between the number of "warm-up" tasks and the resulting cost savings/success rate? How many tasks are required to reach, for example, 80% of the maximum efficiency gain?

2. Have you explored any strategies to mitigate the cold-start problem, such as pre-populating the subroutine library with a set of manually defined or "common-sense" rules from a small seed dataset to bootstrap the learning process?

3. While execution cost (time) is reported, could you provide an analysis of the total inference cost, including the API calls to the LLMs (GPT-4o, GPT-o1) used for planning, subroutine selection, and induction? Does the 49% savings in execution time still hold when these substantial LLM costs are factored in?

4. The system relies heavily on powerful, expensive LLMs (GPT-o1 for induction). Were any ablations performed using smaller, open-source LLMs for any of the components (e.g., subroutine selection) to assess the trade-off between performance and cost/dependency?

5. The paper builds upon the complex CoSTA* framework. Beyond code availability, what are the minimum system requirements (e.g., GPU memory) and estimated engineering effort to set up and run the full FaSTA* pipeline? Is there a containerized environment or a more detailed setup guide to facilitate replication?

6. How reliant is FaSTA* on the specific, pre-defined structures of CoSTA* (the Tool Dependency Graph, Model Description Table, Benchmark Table)? How much manual effort is required to adapt FaSTA* to a new set of AI tools not in the original CoSTA* setup?

7. The paper mentions a verification step for new subroutines, but what happens when a "bad" rule is nonetheless learned and passes verification? Could you provide an example of such a case and its impact on subsequent task performance? How often does the inductive reasoning propose rules that are later found to be flawed?

8. The learned rules seem effective for tasks similar to those in the training distribution. How does FaSTA* perform on a task that requires a genuinely novel combination of tools or operates on image types (e.g., medical imagery, technical diagrams) far outside its experience? Does the fallback to A* search reliably recover quality in these OoD scenarios?

9. Key details for understanding the methodology—such as the full structure of a trace, the exact prompts for inductive reasoning, and the complete Subroutine Rule Table—are in the appendices. Would you consider moving the most critical elements (e.g., the rule table structure, a trace example) to the main paper to make it self-contained?

10. The reviewers note the code is available for review. We strongly recommend you explicitly state in the paper the commitment to make the code and model weights (if applicable) publicly available upon acceptance, to ensure the community can build upon this work.

---

> ### Author Response · Authors · 2025-11-23
> **Response to Reviewer wGeb (1 of 2)**
>
> We thank the reviewer for their detailed feedback and the positive assessment. We address your comments below.
>
> > **W1: The performances are suboptimal initially, requiring a "warm-up" period.**
>
> We have evaluated cold-start in Appendix C.1. With this "suboptimal" initial phase, FaSTA*'s editing quality is identical to the best performed CoSTA* baseline and significantly outperforms other baselines. Furthermore, efficiency starts to improve immediately with the first few learned subroutines. Hence, we do not need a fully populated library to start outperforming the baseline.
>
> > **W2: The entire framework hinges on capable (and often expensive) LLMs creating a dependency and potential cost center.**
>
> In most agentic frameworks, **LLMs are an unavoidable dependency** for planning. In image editing domain, LLM can be a cost-saver. The alternative, A* search, triggers computationally heavy vision tools (e.g., Stable Diffusion, taking seconds on GPUs) tens of times in exploration. An LLM API call is orders of magnitude cheaper and faster than these avoided image generation steps.
>
> > W3: Builds upon the already complex CoSTA* framework and is potentially challenging to reproduce.
>
> CoSTA* has been fully open-sourced. To ensure promising reproducibility of FaSTA*, we have provided:
> 1.  Detailed, simple instructions in the **README** of the attached codebase.
> 2.  A **simple demo notebook** for quick visualization of the pipeline.
> 3.  **Self-contained scripts** for running the benchmark.
> We also explicitly commit to releasing a **containerized version (Docker)** upon acceptance, which will serve as a "plug-and-play" solution handling all dependencies.
>
> > **W4: The success of the "Fast Plan" relies on the generality of the learned symbolic rules. If the learning is bad, the results are properly of bad quality.**
>
> Two verification steps in FaSTA* ensure the generality of the learned symbolic rules:
> 1.  **Offline Verification:** Before induction, every rule is examined on a held-out set (Appendix F.4). Only the rules satisfying the "Net Benefit" criteria are accepted, preventing "bad learning."
> 2.  **Runtime Quality Check (Online):** Even if a suboptimal rule passes the offline verification, FaSTA* employs automatic **VLM Quality Checks** at each step of execution. If the output is poor, the system falls back to the "Slow" A* search.
>
> > **W5: The paper is too short and not self contained.**
>
> We will utilize the additional page allowed in the camera-ready version to move critical elements into the main body of the paper to ensure it is self-contained.
>
> > **Q1: The paper states performance improves with more tasks, but could you provide a concrete curve?**
>
> We have reported this learning curve in **Appendix C.1**.
> *   **Findings:** The improvement is roughly linear in the early stages.
> *   **Conclusion:** It takes approximately **120 tasks** to reach ~80% of the maximum efficiency gain. By 200 tasks, the system is fully "warmed up."
>
> **Table: Learning Curve**
> | Explored Tasks | Avg. Cost (s) | Efficiency Gain |
> | :--- | :--- | :--- |
> | 0 (Cold Start) | 46.80s | 0% (Same as CoSTA*) |
> | 40 | 42.75s | ~9% |
> | 120 | 39.71s | ~15% |
> | 200 (Warmed Up) | 29.25s | **~37%** |
>
> > **Q2: Have you explored any strategies to mitigate the cold-start problem?**
>
> Yes, pre-populating the library is possible **if we utilize prior expert knowledge** about the tools (e.g., hard-coding known effective sequences along with activation conditions). This would allow the system to start with a non-empty rule set, skipping the initial exploration phase.
>
> > **Q3: Could you provide an analysis of the total inference cost, including the API calls? Does the 49% savings hold?**
>
> Yes, the savings hold. Our comparisons with baselines include and exclude exact the same factors to ensure fairness.
> *   **Amortized Overhead:** The only additional cost in FaSTA* is the **Inductive Reasoning** step. As shown in Appendix C.1, the system warms up after ~200 tasks. If we consider a deployment of 2,000 tasks, the overhead of reasoning (on the first 200) is amortized.
> *   **Impact:** Even with this overhead distributed, the savings only drop from **49.3% to ~45.2%**. This is not a significant difference, and the massive savings in exploring different image generation tools vastly outweighs the cost of additional LLM tokens.
>
> > **Q4: Were any ablations performed using smaller, open-source LLMs?**
>
> We tried smaller LLMs (e.g., Qwen 2.5VL 3B) during our preliminary study of this project. Compared to large models, smaller models made more planning mistakes, leading to failed tool calls that wasted GPU time. The cost of these failures can outweigh the savings due to the cheaper LLM API. We will conduct a formal ablation on this and include it in the revised version.

---

> > ### Author Response · Authors · 2025-11-23
> > **Response to Reviewer wGeb (2 of 2)**
> >
> > > **Q5: What are the minimum system requirements? Is there a containerized environment?**
> >
> > All our experiments run on a single **NVIDIA A100**. Inference can run on consumer GPUs (e.g., RTX 3090/4090) with 24GB+ VRAM by offloading unused vision models. We will provide a Docker container and a requirements.txt that handles the complex dependencies for the vision tools (SAM, YOLO, etc.) to facilitate "plug-and-play" replication.
> >
> > > Q6: How reliant is FaSTA* on the specific, pre-defined structures of CoSTA* (the Tool Dependency Graph, Model Description Table, Benchmark Table)? How much manual effort is required to adapt FaSTA* to a new set of AI tools not in the original CoSTA* setup?
> >
> > *   **Benchmark Table (BT)** is needed to provide heuristics for FaSTA*.
> > *   **Tool Dependency Graph (TDG):** This can be **automatically constructed** from the Model Description Table (MDT), requiring no manual effort.
> > *   **New Tools:** Adapting to new tools is very low effort. We only need the tool's reported cost/quality (from its paper), its subtask type, and its Input/Output format (e.g., "mask" vs "image"). This info is easily obtainable, making the framework highly extensible.
> >
> > > **Q7: What happens when a "bad" rule is nonetheless learned?**
> >
> > While our verification steps makes this unlikely, if a bad rule passes, the **Runtime VLM Quality Check** catches the failure during execution. The system then triggers the A* search fallback. This ensures robustness and quality preservation even in the presence of imperfect rules.
> >
> > > **Q8: How does FaSTA perform on a task that requires a genuinely novel combination of tools or OOD image types?**
> >
> > In Out-of-Distribution (OOD) scenarios, the Fast Planner returns "None," and the system defaults to A* search, reliably recovering quality. Crucially, FaSTA* will **record newly explored successful A* traces** and subsequently **learn new rules** for this novel domain, adapting its future behavior Therefore, FaSTA*'s performance floor is exactly that of CoSTA*; it never performs worse than the baseline search on OoD tasks.
> >
> > > **Q9: Moving critical elements to main paper?**
> >
> > Yes, we will move the trace structure and rule table to the main paper in the camera-ready version (one addition page granted).
> >
> > > **Q10: Code and model weights availability.**
> >
> > We explicitly commit to making the **full codebase** and a **containerized environment** publicly available upon acceptance to ensure the community can build upon this work.

---

> > > ### Author Response · Authors · 2025-11-27
> > > **Follow up on the response**
> > >
> > > Dear Reviewer,
> > >
> > >
> > > Thank you again for your time and helpful suggestions. We wanted to follow up to confirm that our responses have fully resolved your concerns. We remain available and are happy to provide any further clarifications should you have additional questions during the remaining discussion period.
> > >
> > >
> > > Sincerely,
> > >
> > > The authors.

---

> > > > ### Comment · Reviewer_wGeb · 2025-11-27
> > > >
> > > > Dear authors,
> > > > Thanks for the clarifications. I will consider them for revising my review.
> > > > Best

---

### Official Review · Reviewer_LasF · 2025-10-31

**Soundness:** 3
**Presentation:** 3
**Contribution:** 3
**Rating:** 6
**Confidence:** 3

**Summary:**

This paper proposes FaSTA* (Fast-Slow Toolpath Agent), an efficient neuro-symbolic agent designed for multi-turn image editing tasks. FaSTA* combines the fast, high-level subtask planning capabilities of large language models (LLMs) with slow, precise tool usage and local A* search.

**Strengths:**

The agent effectively integrates the symbolic reasoning capabilities of large language models (for subroutine mining and high-level planning) with the precision of localized A* search (for low-level tool path optimization), resulting in a robust and efficient system.

**Weaknesses:**

The extraction of subroutines relies on the inductive reasoning capabilities of LLMs, and is represented in the form of symbolic rules (for example, conditions based on object area and mask ratio). The generalization ability and robustness of these rules may be challenged when faced with changes in the toolset or with complex and ambiguous editing tasks.

**Questions:**

Please explain in detail how an LLM performs inductive reasoning to extract symbolic rules, particularly how it determines the quantification conditions in the rules (such as object_area ≤ θ). How are these threshold values determined?

---

> ### Author Response · Authors · 2025-11-23
> **Response to Reviewer LasF (1 of 2)**
>
> Thank you for your detailed feedback! We address your comments below.
>
> > **Q1: The extraction of subroutines relies on the inductive reasoning capabilities of LLMs**
>
> FaSTA* leverages LLMs to **propose hypothetical rules but does not fully relies on them**: the proposed rules are hypotheses and have to go through a careful, rigorous validation process before being accepted. Specifically,
> *   **Empirical Validation:** As defined in **Section 4.2 (Step 4)** and detailed in **Appendix F.4**. Each proposed rule is examined on a **held-out test set** of tasks (described in Appendix K) that were not used for rule extraction.
> *   **Net Benefit Criteria:** The rule is accepted only if it satisfies the **Net Benefit** criteria ($B(\Delta)$), which mathematically ensures that the new rule either reduces cost without degrading quality or improves quality significantly. This filtering avoids "hallucinated" or poorly generalized rules proposed by LLMs.
>
> >**Q2: Robustness of rules may be challenged due to Changes in the Toolset**
>
> FaSTA* is robust and can be directly applied with different variations of toolsets, e.g., upgrading to new versions of tools or addition of new subtasks.
> *   **Adaptive Mechanism:** FaSTA* employs an **Online Learning (Algorithm 2)** rather than a static library of rules. If the toolset changes, some existing rules may fail the quality check by VLMs. In these cases, the system automatically switches to "Slow" A* search to explore the *new* toolset.
> *   **Rule Update:** The newly explored traces enforces the inductive reasoner to discover different trade-offs offered by the new tools. It will then propose updates to the Rule Table, effectively "overwriting" obsolete rules with new ones tailored to the current toolset.
>
> >**Q3:  The generalization ability may be challenged when faced with changes in the toolset or with complex and ambiguous editing tasks**
>
> To handle ambiguous tasks where rigid thresholds fail, we deliberately prompt the LLM to extract **descriptive, semantic rules** rather than hard numerical constraints.
> *   **Hard Thresholds (Brittle):** A rule like `IF object_area < 500 pixels` is brittle because "500 pixels" varies by image resolution.
> *   **Semantic Logic (Robust):** FaSTA* synthesizes rules based on relative descriptors (e.g., `IF object_size == "Small (Relative)"` or `IF background_complexity == "High"`). As detailed in **Appendix L**, our traces include high-level features inferred by VLMs, allowing rules to capture *semantic* nuance (e.g., "don't use inpainting if the object overlaps with text") rather than just raw pixel statistics.

---

> > ### Author Response · Authors · 2025-11-23
> > **Response to Reviewer LasF (2 of 2)**
> >
> > > **Q4: Detailed explanation how an LLM performs inductive reasoning to extract symbolic rules**
> >
> > The inductive reasoning does not involve solving for a mathematical variable; rather, it relies on the LLM's semantic understanding of data distributions to create robust "semantic buckets." The detailed process (outlined in Section 4.2 and Appendix F) operates as follows:
> >
> > **1. Data Logging (Raw Inputs):**
> > First, FaSTA* logs execution traces containing both high-level semantic features (inferred by VLMs, e.g., `background: complex`) and low-level numeric values returned from tools (e.g., `object_size` from YOLO bounding boxes, `mask_confidence` from SAM).
> >
> > **2. Pattern Recognition (Inductive Step):**
> > Periodically, an LLM (GPT-o1) analyzes a batch of recent traces ($T_{recent}$), which includes both successful and failed attempts. The LLM looks for correlations between context features and outcomes.
> > *   *Example:* The LLM observes that `Tool_Sequence_A` succeeded in 10 cases where the object took up 20-50% of the image, but failed in 5 cases where the object was <5% of the image.
> >
> > **3. Determining "Thresholds" (Semantic Bucketing):**
> > Crucially, we do **not** ask the LLM to output a precise threshold (e.g., $\theta = 5.5\%$). Precise numbers often lead to overfitting. Instead, as detailed in our prompt (Appendix R), we explicitly instruct the LLM to formulate rules using **qualitative descriptors**.
> > *   The LLM synthesizes the observation into a rule such as: `IF object_size == "Very Small" THEN Avoid Tool_Sequence_A`.
> > *   The "threshold" is effectively determined by the LLM mapping the raw numeric values from traces to these semantic concepts (`Small`, `Medium`, `Large`), based on its internal world knowledge and the context of the failures.
> >
> > **4. Verification (Empirical Validation):**
> > To ensure these qualitative rules are robust, we employ the verification step (Section 4.2, Step 4).
> > *   Each proposed rule is tested on a held-out set of tasks (Appendix K).
> > *   If the semantic threshold is too loose or too strict, the **Net Benefit Score** will drop.
> > *   Only rules that demonstrate a positive Net Benefit are accepted. This acts as a practical filter, ensuring that the LLM's "subjective" determination of the threshold effectively separates success from failure in practice.

---

> > > ### Author Response · Authors · 2025-11-27
> > > **Follow up on the response**
> > >
> > > Dear Reviewer,
> > >
> > >
> > > Thank you again for your time and helpful suggestions. We wanted to follow up to confirm that our responses have fully resolved your concerns. We remain available and are happy to provide any further clarifications should you have additional questions during the remaining discussion period.
> > >
> > >
> > > Sincerely,
> > >
> > > The authors.

---

### Official Review · Reviewer_KYws · 2025-11-01

**Soundness:** 2
**Presentation:** 3
**Contribution:** 2
**Rating:** 4
**Confidence:** 4

**Summary:**

The paper proposes FaSTA*, a Fast-Slow Toolpath Agent that builds upon CoSTA* (Cost-Sensitive Toolpath Agent) to enhance efficiency in multi-turn image editing tasks. The core idea is to mine frequently used subroutines (i.e., reusable sequences of tool calls) from prior editing experiences using large language model (LLM)-based inductive reasoning. These symbolic subroutines are stored and reused for similar future tasks, allowing the system to perform a “fast plan” by recalling suitable subroutines, with fallback to a slower A*-based search (“slow plan”) if no appropriate rule exists or the quality check fails. Experiments show that FaSTA* reduces computational cost by roughly 49% compared to CoSTA* while maintaining comparable quality.

**Strengths:**

- The idea of leveraging reusable subroutines for planning efficiency is a logical extension of CoSTA*. It introduces an experience-based learning mechanism that mimics human procedural reuse and contributes to more scalable agentic workflows.
- The fast-slow execution strategy and subroutine mining pipeline are well described, supported by detailed diagrams and ablations. The authors conduct a careful comparison with CoSTA* and show meaningful runtime reductions.
- Representing toolpath knowledge as symbolic rules rather than black-box model weights improves transparency and offers a potential bridge between LLM-based reasoning and symbolic planning.

**Weaknesses:**

- Despite its engineering refinement, FaSTA* is largely an incremental enhancement of CoSTA*. The “subroutine mining” mechanism mainly involves clustering or pattern extraction of successful tool sequences using an LLM prompt, without introducing new algorithmic insights or theoretical guarantees. The resulting “fast-slow” hybrid is conceptually intuitive but not methodologically groundbreaking.
- The reported efficiency gains (≈49%) are based on the same dataset and evaluation setting as CoSTA, limiting generalizability.  The comparison to baselines (MagicBrush, GenArtist, CLOVA) is inherited from CoSTA*, with no new external benchmark validation. Human evaluation of image quality lacks standardized metrics or reproducibility details, and visual examples (Fig. 4–8) are mostly qualitative.
- The “subroutine mining” uses CoSTA’s outputs for initialization, meaning FaSTA* effectively benefits from pre-computed search traces, inflating its efficiency advantage. The cold-start performance (on unseen domains or without prior CoSTA* traces) is underexplored and acknowledged as a limitation but not quantified.
- The inductive reasoning step is described as “in-context reinforcement learning,” but in practice it is a rule-extraction procedure with periodic prompt-based updates—no reward modeling or policy improvement is performed. This weakens the conceptual link to ICRL and exaggerates the claimed neurosymbolic contribution.

**Questions:**

NA

---

> ### Author Response · Authors · 2025-11-23
> **Response to Reviewer KYws (1 of 2)**
>
> Thank you for your detailed feedback! We address your comments below.
>
> > Q1: FaSTA* is largely an incremental enhancement of CoSTA*. The “subroutine mining” mechanism mainly involves clustering or pattern extraction without introducing new algorithmic insights.
>
> **FaSTA\* addresses a different problem as CoSTA\*,** i.e., how to learn reusable subroutines from previous experiences and maintain a dynamic memory of them to improve future editing tasks, while CoSTA* is an LLM-aided A\* search approach without any learning. Fig. 1 shows the outcome and gain of learning in FaSTA\*, which cannot be achieved by non-learning CoSTA\*. In FaSTA\*, CoSTA\* performs as a tool to be called by the agent for further exploration when none of the learned subroutines applies.
>
> **Novelty of FaSTA\*:** it firstly introduces symbolic subroutine learning to image editing agents, improving efficiency on challenging multi-turn image editing tasks. Its algorithmic novelty, as illustrated in **Appendix I.1**, is not limited to simple pattern extraction or clustering.
>
> For example, the inductive reasoning of abstract subroutines is critical to the improvement achieved by FaSTA*. This can be verified by comparing FaSTA* with a **"Direct Caching" baseline**, which removes inductive reasoning from FaSTA* and direclty reuse toolpaths retrieved by clustering or simple pattern matching.
>
> *   Direct Caching achieves a **95% fallback rate** (i.e., it failed to apply cached paths to new tasks 95% of the time) because exact context matches in pixel/feature space do not hold in complex image editing.
> *   FaSTA\* achieved a **9% fallback rate**. It employs **inductive reasoning** to synthesize **symbolic, conditional logic** (e.g., abstracting that `object_size` is the critical failure factor for a specific tool, while `color` is irrelevant).
>
> **This 86% performance gap justifies the importance of inductive reasoning in FaSTA\*,** which is a novelty beyond clustering or pattern matching. It leads to a **generalizable editing policy** that can handle unseen tasks where simple toolpath retrieval fails.
>
>
> > **Q2: Human evaluation of image quality lacks standardized metrics or reproducibility details. The reported efficiency gains are based on the same dataset and evaluation setting as CoSTA, limiting generalizability.**
>
> The human evaluation is rigorous and standardized with promising reproducibility. **Appendix J (Table 12)** lists a comprehensive rubric for human evaluators to score partial success (e.g., precise penalties for artifacts vs. semantic alignment), which ensures consistency.
>
> Our evaluation was conducted with a panel of **five human evaluators** from diverse professional backgrounds. They were tasked with rating outputs based on this detailed rubric, and we averaged their ratings as the final results. The **variance in scores among evaluators is 0.07 and notably low**, indicating strong inter-evaluato consistency and that the detailed criteria successfully mitigated subjectivity. To further improve transparency, we included 12 distinct qualitative examples in the paper for readers to visually assess the performance.
>
> To address the generalizability concern, we also conducted additional experiments on a subset of **Complex-Edit**, an open-source image-editing benchmark with up to 8 turns. As shown in the table below, FaSTA* maintains its efficiency advantages on this independent benchmark with a decrease of around 30% in costs with marginal change in quality.
>
> | Task Complexity | Metric | CoSTA* | FaSTA* (Ours) |
> | :--- | :--- | :--- | :--- |
> | **1-3 Subtasks** | Cost (s) | 46.75 | **35.87** |
> | | Quality (Human Eval) | 0.88 | 0.86 |
> | **4-5 Subtasks** | Cost (s) | 77.25 | **54.17** |
> | | Quality (Human Eval) | 0.90 | 0.89 |
> | **6-8 Subtasks** | Cost (s) | 105.60 | **73.20** |
> | | Quality (Human Eval) | 0.90 | 0.88 |
> | **Overall** | **Avg. Cost (s)** | **78.27** | **55.12** |
> | | **Avg. Quality (Human Eval)** | **0.89** | **0.87** |

---

> > ### Author Response · Authors · 2025-11-23
> > **Response to Reviewer KYws (2 of 2)**
> >
> > > **Q3: The “subroutine mining” uses CoSTA’s outputs for initialization. The cold-start performance is underexplored and acknowledged as a limitation but not quantified.**
> >
> > FaSTA* does not have to rely on pre-computed CoSTA\* traces; it can explore traces in an online manner and dynamically learn subroutines on the fly. When initialized with zero prior experience, FaSTA\* simply defaults to the "slow" planner (the CoSTA* tool) and explore enough experience to induce rules.
> >
> > We have explicitly quantified this "Cold Start" learning curve in **Appendix C.1**. The table below illustrates "Cold Start" agent's behavior from initialization:
> >
> > **Table: Cold Start Performance Analysis**
> > | Tasks Explored | Avg Cost (s) | Performance Note |
> > | :--- | :--- | :- |
> > | 0 (Cold Start) | 46.8 | Identical to CoSTA* baseline |
> > | 40 | 42.8 | Initial efficiency gains appear |
> > | 120 | 39.7 | Achieves ~80% of max efficiency |
> > | 200 (Warm) | 29.3 | Reaches optimal efficiency |
> >
> > This decrease in cost demonstrates the efficiency advantage obtained by rapidly online learning of FaSTA*, which is not an artifact of pre-computation.
> >
> > > **Q4: The inductive reasoning step is described as “in-context reinforcement learning,” but in practice it is a rule-extraction procedure. no reward modeling or policy improvement is performed.**
> >
> > FaSTA* dynamically improves a bank of reusable subroutines (which composes the **agent policy**) based on continuously explored trajectories (the **context** of inductive reasoning) and their achieved Net Benefit Scores (the **reward**). This aligns with the core setting of In-Context Reinforcement Learning (ICRL), where an agent improves its policy given explored trajectories and feedback, without updating model weights. Specifically,
> >
> > 1.  **Policy Learning:** The "Subroutine Rule Table" functions as an interpretable, evolving **editing policy**. Unlike static caching, this policy generalizes across multiple trajectories.
> > 2.  **Reward Signal:** the **Net Benefit Score** (balancing cost vs. quality, Section 4.2) and the **VLM-based quality validation** serve as the reward signal.
> > 3.  **Policy Improvement:** The "Update" step of the inductive reasoner utilizes this reward to explicitly accept, reject, or refine the symbolic rules. It improves the agent policy to maximize future rewards.

---

> > > ### Author Response · Authors · 2025-11-27
> > > **Follow up on the response**
> > >
> > > Dear Reviewer,
> > >
> > >
> > > Thank you again for your time and helpful suggestions. We wanted to follow up to confirm that our responses have fully resolved your concerns. We remain available and are happy to provide any further clarifications should you have additional questions during the remaining discussion period.
> > >
> > >
> > > Sincerely,
> > >
> > > The authors.

---

### Author Response · Authors · 2025-12-03
**General Response: Summary of Clarifications and Revisions**

Dear Reviewers and Area Chair,

We deeply appreciate the insightful and constructive comments provided by all reviewers.

We are encouraged by the reviewers' recognition of **FaSTA*** as an "**intuitive and powerful**" framework (Reviewer wGeb) that offers a "**logical extension**" to agentic workflows with "**meaningful runtime reductions**" (Reviewer KYws). FaSTA* addresses the critical bottleneck of inference latency in multi-turn image editing by bridging the gap between fast LLM reasoning and slow, robust search.

To address the shared concerns regarding novelty, fairness, and generalizability, we have provided detailed clarifications and additional experimental results in our individual responses. We have incorporated all new results and clarifications into the revised manuscript, with major updates highlighted in $\color{blue}{\text{blue}}$.

**Summary of Main Concerns of reviewers and our Responses:**

*   **Novelty (Reviewers `KYws`, `39Yj`): Inductive Reasoning $\neq$ RAG/Caching**
    We clarified that our contribution is the synthesis of generalized symbolic policies, not simple retrieval of raw experiences. As demonstrated in **Appendix I.1**, we compared FaSTA* against a "Direct Caching" (RAG) baseline. The RAG baseline failed with a **95% fallback rate** because exact context matches are rare. In contrast, FaSTA*'s inductive reasoning achieved a **9% fallback rate**. This **86% performance gap** demonstrates our method learns robust conditional logic rules rather than merely recalling past solutions.

*   **Fair Comparisons on Identical Toolsets (Reviewer `39Yj`): Our method still outperforms baselines when constrained to the same toolsets as baselines.**
    To ensure the performance gains stem from our fast-slow planning with symbolic rule learning rather than a more comprehensive toolset, we presented strict ablation studies in our response to Reviewer `39Yj`. Even when FaSTA* is restricted to **exactly the same toolsets and subtasks** as the baselines (VisProg, GenArtist, CLOVA), it significantly outperforms them, confirming the value of the **Fast-Slow architecture and online learning**.

*   **Do cold-start or LLM overhead negate efficiency gains? (Reviewers `KYws`, `wGeb`): FaSTA*** **starts with identical performance to the SOTA baseline and rapidly gains efficiency, with net savings holding even after accounting for LLM costs.**
    We highlighted the learning curve analysis of FaSTA* with "cold start" in **Appendix C.1**. FaSTA*'s quality and efficiency begin identical to the best baseline CoSTA*. Then, its cost rapidly decreases with more subroutines learned for the fast planning, and it reaches **~80% of its maximum efficiency within just 120 tasks**. We also clarified that the **~50% reduction in total inference time** already takes LLM overhead into account: the fast planning saves cost for 10-20 expensive image generation steps, which vastly outweighs the cost of LLM reasoning.

*   **Generalizability to More Benchmarks (Reviewers `KYws`, `LasF`): New experiments on Complex-Edit benchmark confirm a ~30% efficiency gain with comparable quality.**
    Beyond the primary benchmark, we conducted new experiments on the Complex-Edit benchmark during the rebuttal phase. FaSTA* maintained its efficiency advantage (~30% cost reduction) with comparable quality on this independent dataset, demonstrating robustness and consistent advantage across different data distributions.

*   **Reproducibility**
    We have committed to releasing the full codebase, the curated benchmark dataset, and a **containerized (Docker) environment** to ensure easy reproducibility of our results.

We believe these clarifications directly address the core concerns raised. We remain committed to incorporating these details into the final manuscript and welcome any further discussion.

Best regards,

Authors

---

### Meta-Review · Area_Chair_Vigv · 2026-01-05

**Summary:**

The paper proposes FaSTA*, a Fast-Slow Toolpath Agent that builds upon CoSTA* (Cost-Sensitive Toolpath Agent) to enhance efficiency in multi-turn image editing tasks. The core idea is to mine frequently used subroutines (i.e., reusable sequences of tool calls) from prior editing experiences using large language model (LLM)-based inductive reasoning. These symbolic subroutines are stored and reused for similar future tasks, allowing the system to perform a “fast plan” by recalling suitable subroutines, with fallback to a slower A*-based search (“slow plan”) if no appropriate rule exists or the quality check fails. Experiments show that FaSTA* reduces computational cost by roughly 49% compared to prior CoSTA* while maintaining comparable quality.


The authors provide detailed clarification, comparison, and ablation studies,  which address the Reviewer KYws and Reviewer 39Yj's key concerns about "novelty" , motivation, and effectiveness.

**Reviewer Concerns:**

The reviewer KYws is concerned about the incremental contribution compared with prior work CoSTA*, and the limiting generalizability, as well as the limited advantage of the “subroutine mining” that may benefit from CoSTA’s outputs as initialization.


The Reviewer 39Yj is also concerned about the Limited Technical Contribution, Potentially Inequitable Experimental Comparison, as well as Questionable Problem Framing and Motivation.


The Reviewer wGeb is also concerned about requiring a "warm-up" period of task execution using the CoSTA’s outputs as initialization before the subroutine library becomes effective.

The authors provide additional clarification, comparison, and ablation studies.

Particularly, regarding the concern about the "novelty", the authors highlight the contribution of proposing symbolic subroutine learning in mage editing agents.   In addition, the authors provide a comparison with "Direct Caching",  which demonstrates the advantage of the inductive reasoning of abstract subroutines in FaSTA*.

Regarding the motivation, the authors highlight the importance of reducing costs with supporting references.

 I think the Reviewer KYws and Reviewer 39Yj's concerns may be addressed.

**Reviewer Scores:**

This is a borderline paper with scores 4, 4, 6, 8.

The authors provide additional clarification, comparison, and ablation studies,  which may address the Reviewer KYws and Reviewer 39Yj's key concerns about "novelty" , motivation, and effectiveness.

---

### Decision · Program_Chairs · 2026-01-26

Accept (Poster)